# An object-based and Lagrangian view on an intense hailstorm day in Switzerland as represented in COSMO-1E ensemble hindcast simulations

Killian P. Brennan[1], Michael Sprenger[1], André Walser[2,†], Marco Arpagaus[2], and Heini Wernli[1]

[1]Institute for Atmospheric and Climate Science, ETH Zürich, Zurich, Switzerland,
[2]Federal Office of Meteorology and Climatology MeteoSwiss, Zurich, Switzerland
[†]deceased, 19 February 2025

**Correspondence:** Killian P. Brennan (killian.brennan@env.ethz.ch)

**Abstract.** On 28 June 2021, Switzerland experienced the passage of several formidable hailstorms, navigating its complex terrain. They unleashed severe hailstones measuring up to 9 cm in diameter. We present a comprehensive model-based case study to unravel the complex processes involved in the genesis, intensification, and dissipation of this impactful weather event. To this end, we investigate ensemble hindcast simulations using the COSMO-1E numerical weather prediction model that includes the HAILCAST hail growth parameterization. A tracking algorithm is introduced that facilitates the object-based analysis of the simulated hailstorms. We show that COSMO-1E with HAILCAST produces realistic storm tracks and lifespans that are in good agreement with radar observations. By scrutinizing the storm's evolution across various phases, particularly during intensification, the study conducts a storm-relative analysis of 100 hailstorms simulated on this day by the 11-member ensemble with lifetimes of > 2.5 h. Furthermore, the investigation utilizes air parcel trajectories initiated along the hail track to analyze the inflow of air sustaining the storm updraft. This exploration identifies specific low-level flow patterns that contribute to the storm's vigor. The most important findings from this detailed, and combined Eulerian and Lagrangian hailstorm analyses are (i) the hailstorms propagate towards a local CAPE maxima; (ii) hailfall is directly followed by intense precipitation; (iii) the inflow detaches from the lowest 800 m up to 30 min before storm dissipation; and (iv) Lagrangian analysis reveals a distinct time lag between rain, cloud condensate and graupel concentrations along the inflowing trajectories.

## 1 Introduction

Severe convective storms offer some of the most intense and impressive weather on our planet. Hailstorms dramatically display the forces of nature at play inside so-called supercells. Supercell thunderstorms are a distinct form of cumulus structure, representing the most severe category of thunderstorms (Schmid et al., 1997; Markowski and Richardson, 2010; Bluestein, 2013; Houze, 2014). Unlike their counterparts, i.e., single and multi-cell storms, supercells are relatively rare but notorious for generating the most intense hailstorms and powerful tornadoes (Graf et al., 2011). Distinguished by their towering vertical reach spanning the troposphere and rotating updraft, supercells surpass the typical scale of single-cell storms. Within them, a singular storm-scale circulation dominates (Markowski and Richardson, 2010; Houze, 2014). Thunderstorms with strong

enough updrafts can form hail, as hail embryos collect mass when they are lofted into the region with supercooled liquid water from the storm by strong updrafts (Pruppacher and Klett, 2010; Allen et al., 2020). Hail ranks among the costliest atmospheric phenomena at mid-latitudes (Changnon, 1999; Crompton and McAneney, 2008). In Switzerland, hailstorms are a frequent phenomenon during the convective season (Houze et al., 1993; Nisi et al., 2016), causing extensive damage to agriculture, buildings, and cars (BAFU, 2016). Summer 2021 was the most active hail season in Swiss records, resulting in various studies that collected and analyzed radar, hail sensor, and crowd-sourced observations (Kopp et al., 2022), measured the hail size distribution by drone (Lainer et al., 2024), while other studies modeled hail damage to agriculture and infrastructure in this season (Portmann et al., 2024; Schmid et al., 2024). The number of days with hail in Switzerland has increased significantly in the last 50 years (Wilhelm et al., 2024), further highlighting the importance of hail research in this region.

The potential for extensive damage illustrates the need to improve forecasts of hailstorms and advance our understanding of the meteorological processes associated with hailstorms. Using parameters or proxies such as convective available potential energy (CAPE), vertical wind shear, and storm-relative helicity (SRH), predictions about the probability of occurrence and intensity of convective storms can be made (i.e., Ulbrich and Atlas, 1982; Marcos et al., 2021). At short lead times, radar-based nowcasting serves to inform about hail occurrence, where the future state of hailstorms is extrapolated from current radar observations and movement vectors of convective cells (e.g., Hering et al., 2004; Trefalt et al., 2023). Predictions with lead times beyond 1–3 h must rely on numerical weather prediction (NWP) models (e.g., Sun et al., 2014; Nerini et al., 2019). In addition to radar-based investigations, high-resolution numerical weather prediction models with explicitly simulated deep convection allow novel insights into the physics of convective storms. As there is inherent uncertainty in the prediction of atmospheric processes (Lorenz, 1963) and especially the position and intensity of mesoscale storms, ensemble models were developed to address this issue and assess the uncertainty of forecasts.

The challenges in predicting hail stem from two main factors: hail formation occurs within intense convective storms, where the prediction of the time and location of the convective triggering, as well as the resulting intensity, is influenced by complex mesoscale processes (e.g., Ducrocq et al., 2008; Barthlott and Kalthoff, 2011; Barthlott and Barrett, 2020). Additionally, the microphysical processes governing hail formation are intricate and not adequately represented in most operational microphysics parameterization schemes (e.g., Adams-Selin and Ziegler, 2016; Brimelow, 2018; Allen et al., 2020). An effective strategy to tackle these challenges is to conduct high-resolution ensemble forecasts (Sect. 2.1) and diagnose the occurrence of hail with a suitable parameterization scheme (Sect. 2.1.1).

In this study, we utilize a single convection-permitting ensemble hindcast simulation, performed with the model COSMO-1E (Klasa et al., 2018), to examine a hail day that severely impacted Switzerland in the summer of 2021. Hail formation is not explicitly simulated, as done in previous studies using the COSMO model with an extended cloud microphysics parameterization including a hail category (e.g., Seifert and Beheng, 2006; Noppel et al., 2010), but instead hail formation is diagnosed during model integration using the HAILCAST parameterization (Adams-Selin and Ziegler, 2016; Adams-Selin et al., 2019), see Sect. 2.1.1. With this case study we contribute to the body of literature on severe weather events in Central Europe and beyond, making use of a high-resolution ensemble simulation and sophisticated diagnostics. Several earlier studies (Schiesser et al., 1995; Trefalt et al., 2018; Rigo et al., 2022; Bechis et al., 2022) analyzed severe hailstorms in complex topography based

on radar observations. Barras et al. (2021) investigated the temporal clustering of hail days in Switzerland and Mohr et al. (2020) considered the role of large-scale dynamics in a sequence of severe thunderstorms in Europe. The specific objectives of our ensemble simulation-based investigation are to (i) explore the physical processes and environmental conditions driving the storm's evolution, (ii) analyze the low-level inflow of air into the hailstorm and the evolution throughout its life cycle, and (iii) assess the storm structure throughout its development. The use of the ensemble simulations provides us with a larger set of physically consistent storm tracks (compared to single deterministic simulations), enabling a more robust interpretation of the relevant physical processes.

Previous studies have investigated atmospheric parameters and environments that influence hailstone size. Analysis in these studies is generally based on observational data on hail in a given region and analysis of reanalysis data in that same region (e.g., Taszarek et al., 2017, 2020; Zhou et al., 2021; Calvo-Sancho et al., 2022). A similar analysis can be performed in convection-permitting NWP models, where the environment is assessed in the immediate vicinity of a storm as it moves through the domain (e.g., Prein et al., 2017; Lin and Kumjian, 2022). Furthermore, rather than investigating the Eulerian neighborhood of a convective storm, backward trajectories can be employed to specifically investigate the origin of air feeding the storm updraft, as shown in recent publications, where the low-level inflow and vorticity sources of supercells in idealized simulations were investigated in the Lagrangian framework (Gowan et al., 2021; Lin and Kumjian, 2022; Coffer et al., 2023). The Lagrangian perspective was recently applied to the research of Foehn phenomena in the Swiss Alps (Jansing et al., 2024). In addition to an Eulerian framework of analyzing atmospheric processes, such a combined object-based and Lagrangian approach — e.g., by investigating backward trajectories from the core updraft region along simulated storm tracks — offers complementary insights into mechanisms at play inside storms. Technically, such an approach therefore requires a specific tracking algorithm for tracking hailstorms in high-resolution simulation output, and a trajectory tool to study the airflow through the core updraft regions. For the latter, we use an established trajectory tool, whereas, for the former, we introduce a storm-tracking scheme specifically developed for our high-resolution model output (see Sect. 2).

Despite foundational studies like Browning and Ludlam (1962), Browning (1964), and Davies-Jones (1984) that investigated airflow patterns within severe local storms, recent literature predominantly emphasizes hailstone trajectories over air parcel trajectories in convective systems. Browning (1977) and Rotunno and Klemp (1982) explored storm propagation mechanisms, while Rotunno (1993) analyzed the three-dimensional airflow structure of supercell thunderstorms. Gaudet and Cotton (2006) focused on supercell and mesocyclone evolution, highlighting the critical correlation between vertical velocity and vertical vorticity. More recent studies have primarily addressed hail growth trajectories rather than air parcel movements. Dennis and Kumjian (2017) investigated the impact of vertical wind shear on hail growth in simulated supercells, referring to hail embryo paths as "pseudotrajectories." Similarly, Kumjian and Lombardo (2020) developed a hail growth trajectory model to explore environmental controls on hail size but did not examine air parcel trajectories. Lin and Kumjian (2022) is one of the few studies that calculated air parcel trajectories, focusing on the influences of CAPE on hail production in simulated supercell storms. However, their trajectory calculations were performed on an idealized, steady-state storm-centered composite with a 1-second time step, which differs significantly from our 5-minute time step time-evolving approach. Others, such as Horner

and Gryspeerdt (2023), considered trajectories in the context of large-scale tropical convection and cirrus outflows. Given that most trajectory analyses in convective storms concentrate on hailstones rather than air parcels, our work offers novel insights into storm dynamics and presents challenges in embedding our results in those from the existing literature.

While this study employs a newly developed hailstorm tracking method tailored specifically for our simulations—an essential tool for our life cycle composite analyses—it is not our intention to demonstrate that this tracking approach is superior to existing methods. We also acknowledge that validating an ensemble simulation model with a single case study is fundamentally impossible; therefore, we cannot first validate our model before investigating the simulated hailstorm tracks. Additionally, we recognize that our analysis focuses on a single, albeit exceptionally intense, hailstorm event in Switzerland, which may limit the generalizability of our conclusions. Nevertheless, we consider our approach both original and important. By utilizing the operational ensemble simulations from MeteoSwiss for this hailstorm day as a "free set of sensitivity experiments", and treating each ensemble member as physically consistent and equally likely, we examine hailstorm tracks across all ensemble members. This enhances the statistical robustness of our findings and allows us to derive more reliable insights into aspects such as the environmental conditions influencing hailstorm tracks.

A brief discussion of the synoptic situation introduces the case study (Sect. 3.1), followed by a description of the storm tracks (Sect. 3.2). The storm-centered perspective is then applied to investigate the spatial and temporal structure of a selected storm and a composite of all long-lived storms simulated by the ensemble in the case study period (Sect. 4). In Sect. 5, the inflow of air into the storm and its implications for the storm's evolution are investigated. Furthermore, the role of topographical features in influencing the various stages of the storms will be examined (Sect. 4). Overall, the study revolves around unraveling the complexities of hailstorm development, providing insights into the storm's life cycle, and improving our understanding of hailstorms in numerical weather simulations.

## 2 Methods and data

This section introduces the data and methods used in this study. The numerical weather model COSMO-1E and the HAILCAST parameterization for hail diameter on the ground are described in Sect. 2.1 and Sect. 2.1.1, respectively. The tracking algorithm used to identify and track the storms in the COSMO-1E simulations is described in Sect. 2.2. The ECMWF global ERA5 reanalysis (Hersbach et al., 2020) was used in this study to characterize the large-scale atmospheric conditions and provided boundary conditions for the regional weather simulations with COSMO.

### 2.1 COSMO simulations

The COSMO model is a non-hydrostatic limited-area NWP model. The governing equations describing compressible flow in a moist atmosphere are solved on a rotated-pole-structured grid with hybrid terrain-following height coordinates (Steppeler et al., 2003). Although designed and optimized for operational NWP, the COSMO model is also extensively utilized in scientific applications on the meso-$\beta$ and meso-$\gamma$ scale. The COSMO model is most suitable for forecasts at a convection-permitting scale (Baldauf et al., 2011). Parameterizations represent unresolved subgrid-scale physical processes, including a single-moment

bulk microphysics scheme (Lin et al., 1983) with five species (cloud water, cloud ice, rain, snow, and graupel) and schemes for shallow convection, boundary layer turbulence, radiation, and land-surface processes. The turbulence parameterization is adapted from Mellor and Yamada (1982) with a prognostic equation for the turbulence kinetic energy, including effects from subgrid-scale condensation and evaporation. It is applied to the bottom boundary of the atmospheric model to calculate surface-layer fluxes (Baldauf et al., 2011). The parameterization of radiation follows the scheme of Ritter and Geleyn (1992). Note that hail is not explicitly simulated as a species in the microphysical parameterization but is calculated diagnostically (Sect. 2.1.1).

COSMO-1E is the operational ensemble model formerly used at MeteoSwiss (Klasa et al., 2018, 2019). COSMO-1E features 11 ensemble members, which are calculated every 3 h for the next 33 h. The ensemble members have varying initial and boundary conditions, as well as stochastically perturbed parameterization tendencies. The horizontal grid size is 1.1 km with $1170 \times 786$ grid-points, covering the entire Alpine region (Fig. 1a). Vertically it extends through 80 model layers to an altitude of 22 km. The model runs with a time step of 10 s, and atmospheric fields relevant to our application are written to disk every 5 min. For this study, a COSMO-1E ensemble hindcast was initialized at 06:00 UTC on 28 June 2021 in its operational setup, just with more frequent output. The simulation output amounts to $\sim 17$ TB of data.

### 2.1.1 HAILCAST

HAILCAST is a diagnostic, physics-based hail growth parameterization. It consists of a one-dimensional, steady-state cloud model which is coupled with a time-dependent hail growth model (Brimelow et al., 2002; Brimelow, 1999; Jewell and Brimelow, 2009; Poolman, 1992). HAILCAST estimates hail size (maximum, mean, and standard deviation of the diameter) expected at the ground. CAM-HAILCAST, used herein, embeds a pseudo-Lagrangian 1D hail growth model into a convection-permitting model (Adams-Selin and Ziegler, 2016; Adams-Selin et al., 2019). As we only use this one version in this study, we will refer to it as HAILCAST. Hailstone growth is modeled through liquid water accretion, ice particle collection, condensation, and sublimation. The hailstone temperature is explicitly calculated, determining wet and dry growth regimes. As HAILCAST is a one-dimensional model, horizontal advection of hail across grid-points is not accounted for, however, the horizontal motion of hailstones across the updraft is parameterized by adding a time-dependent updraft multiplier term. Multiple initial embryo sizes are injected at various temperatures into the updraft and their size is tracked along their vertical path through the convective cell. However, for this study, only the hail size yielded by the largest, 10 mm embryo was considered. In addition, HAILCAST features include variable hail density, rime soaking, temperature-dependent ice collection efficiency, liquid water shedding, and enhanced melting during collision with $> 0°C$ water (Adams-Selin and Ziegler, 2016).

### 2.2 Storm tracking algorithm

Hailstorms can be relatively small atmospheric features that may move at high velocities, which poses a complication to tracking tools, as such features might not overlap spatially between two model output time steps. In the tracking algorithm used in this study, hailstorms are identified as two-dimensional objects formed by grid-points where a parameter (e.g., max.

hail size) exceeds a certain threshold. An adaptive threshold allows small, high-intensity storms to be separated in a larger, mesoscale convective system. Further, an adaptive threshold allows storms to be tracked in their developing, mature, and dissipating stages, which might have very different intensities.

None of the tracking implementations available fulfill all of the tracking requirements needed for our application, such as dynamical tracking, adaptive threshold, and accounting of splitting and merging. We therefore present a novel tool developed to identify and track features associated with convective storms. It is optimized to track small, fast-moving objects in two-dimensional fields of limited-area atmospheric model simulations with high temporal and spatial resolution. In essence, the algorithm detects objects with a corresponding mask based on various filtering criteria, such as area, intensity, and distance, and uses the last known object propagation vector to inform the search area during the next tracking time step. The tracking algorithm also accounts for splitting and merging objects and can solve complex scenarios involving multiple objects with non-trivial evolution pathways. For a detailed description of the tracking algorithm, please refer to Appendix A. In the following, only tracking parameters and details specific to this study are listed. The tracking was performed for updrafts identified in the vertical wind field $w$ on model level 25, which corresponds to an average altitude of $z = 7.5\,\mathrm{km}$, or pressure of $400\,\mathrm{hPa}$. Features with $w > 5\,\mathrm{m\,s^{-1}}$ were tracked, while features with a prominence exceeding $20\,\mathrm{m\,s^{-1}}$ and with maxima separated by more than 10 grid-points were divided using a watershed algorithm. The area threshold was set to 5 grid-points, while the storm mask was inflated by 4 grid-points using binary dilation, which applies a circular disk kernel to the storm mask to expand the borders of the mask.

## 3 Overview of case study

The selection of the case study date can be justified by the extensive damages that occurred on 28 June 2021. Large areas of the Swiss plateau were impacted by damaging hail, while hailstones with diameters in excess of $9\,\mathrm{cm}$ wreaked havoc in select villages in the Alpine foothills. In fact, the largest area within Switzerland affected by severe hail since 2002 was recorded on this day, with return periods locally exceeding 100 years (Kopp et al., 2022). More than $10\,000$ crowd-sourced reports were collected on that day, which represents the highest daily number on record. At the time, reports collected in June and July 2021 accounted for half of all reports collected since the introduction of the reporting function more than 5 years earlier (Kopp et al., 2022). Insured building damage in 4 cantons alone accounts for more than 400 million Swiss Francs (CHF), with more than 1000 heavily damaged buildings with $> 100\,000$ CHF damage each (Schmid et al., 2024).

### 3.1 Synoptic situation

On 28 June 2021, analysis of the upper-level flow situation over western and central Europe reveals a prominent potential vorticity ($PV$) cutoff over western France (Fig. 1a). It moved eastwards from the French Atlantic coast towards the main Alpine crest and thereby brought a cold air anomaly aloft while advecting warm and moisture-laden Mediterranean air towards the Swiss plateau at low levels. The $PV$ cutoff originated from a $PV$ streamer over the British Isles that formed four days earlier. Besides the presence of a shallow surface cyclone over north-western France, the pressure distribution across central

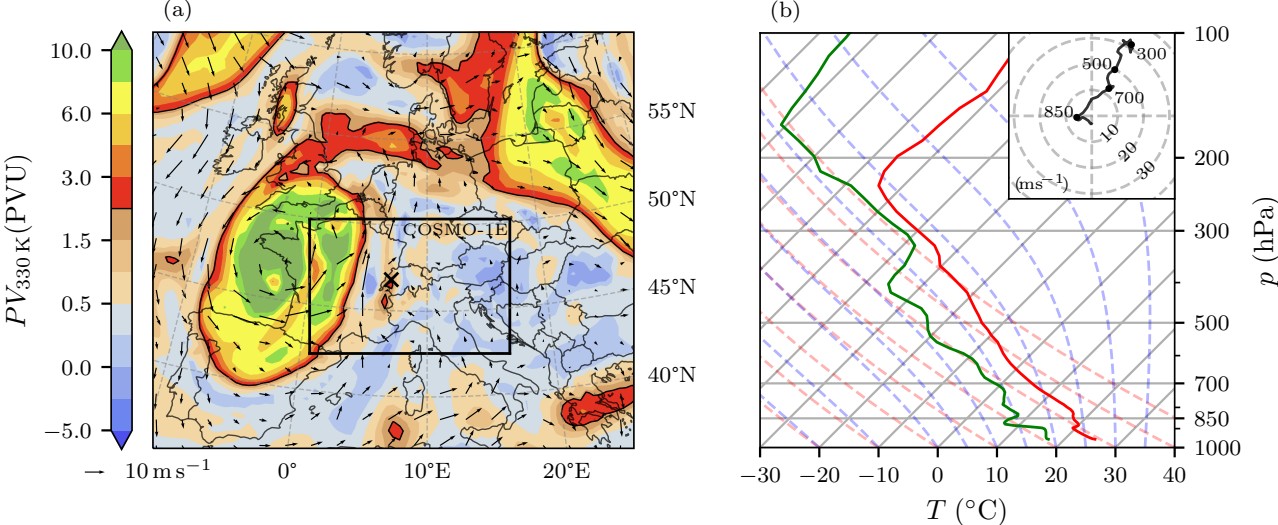

**Figure 1.** Synoptic situation at 12:00 UTC on 28 June 2021. (**a**) Potential vorticity on the 300 K isentrope (colors, in PVU; the black contour denotes the dynamical tropopause at 2 PVU), the quivers show horizontal wind at 850 hPa, based on ERA5. The × marker locates the position of Payerne. The COSMO-1E domain is also delineated (black rectangle). (**b**) Skew-T Log-P diagram of the vertical profile in Payerne (from COSMO-1E, member 0). The ambient temperature and dew point profiles are shown in red and green solid lines, respectively, while the red and blue dashed lines represent the dry and moist adiabats. The hodograph displays the $u$ and $v$ wind components from the surface to 200 hPa.

Europe was flat (i.e. only weak horizontal gradients). In Switzerland, $T_{2m}$ approached $30°$C in the pre-storm environment, with dew-point temperature around $20°$C. Widespread CAPE $> 2000\,\mathrm{J\,kg^{-1}}$ as well as some CIN were present throughout the domain, accompanied by 0–6 km directional shear larger than $25\,\mathrm{m\,s^{-1}}$, yielding an atmosphere primed for organized convection (Fig. 1b). The vertical profile also reveals a moderate level of moisture throughout the hail growth layer (HGL, extending from $0°$C to $-38°$C levels). It should be noted that the HGL is even moister in the measured balloon sounding from Payerne at 12:00 UTC (not shown). The measured profile also features a more pronounced low-level inversion than the simulated profile. This inversion inhibits the premature destruction of CAPE through unorganized convection and allows for further accumulation of heat and moisture in the boundary layer throughout the day. As a result, an intense and long-lived mesoscale convective system originating in western Switzerland moved along the main Alpine crest in a north-easterly direction throughout the day, as further discussed below. More details about the weather situation on this day and radar imagery can be found in Kopp et al. (2022).

## 3.2 Storm tracks

Storm tracks were determined in all ensemble members using the algorithm described in Sect. 2.2. Although there are 6611 storm tracks across all members with a lifespan $> 15$ min, only 124 storms have a lifespan $> 2.5$ h, of which 100 storms feature updraft velocities $> 25$ m s$^{-1}$ at 400 hPa. Storm lifespans and maximum storm areas both follow a log-normal distribution. The ensemble members are generally well in agreement in terms of the produced storm lifetimes and areas, although the ensemble spread is larger at the tails of the distribution (Fig. 2). The distribution of the simulated storm lifespans aligns very well with the observed storm tracks from the radar. However, here it must be kept in mind that for the observations and the simulation, two different tracking algorithms on two different fields were used and thus direct comparisons are only meaningful to a limited extent. For bins of life expectancy below 1 h, the ensemble spread, although narrow, contains the observed lifespan prevalence. The number of longer-lived storms with lifespans $> 2$ h is underestimated by a factor of $\sim 2$. The agreement between the simulated and observed storm areas is lower, especially with respect to the smallest storm areas (Fig. 2b). These differences potentially stem from the different tracking algorithms used.

Although most of the long-lived tracks are centered on the Napf region (47°N, 8°E, Fig. 3), there are also some isolated tracks upstream and downstream of this region. Interestingly, almost all simulated storms occur north of the Alpine crest. Compared to the measured tracks, the simulation misses some of the South-Alpine storms, while overproducing storms further downstream in the Black Forest region. But overall, focusing on the severe hail event in northern Switzerland, this analysis has shown that the characteristics of the simulated storm tracks are comparable to the measured tracks on the case study day.

Next, we consider a subset of all storm tracks to be investigated in detail in Sections 4 and 5. Specifically, 2.5 h was the selected storm lifespan threshold for a detailed analysis of the storm inflow and structure, using storm-centered and Lagrangian perspectives. This lifespan threshold gives a good compromise of long enough lifespans to investigate the storm's life cycle while still allowing for robust statistics thanks to the use of an ensemble simulation. Furthermore, only long-lived storms that achieved updraft velocities greater than 25 m s$^{-1}$ at least once within their lifetime were considered ($n = 100$). Although the storms selected using these criteria only account for 1.7% of the tracked storms, 12.6% of the storms that exhibit $w > 25$ m s$^{-1}$ within their evolution feature lifespans $> 2.5$ h, meaning that the investigation of long-lived storms favorably covers intense storms (Fig. 2c). To facilitate the investigation of storm initiation, the initial time step of each storm is extrapolated backward in time by 0.5 h using the mean lifetime storm propagation vector.

Preceding the analysis of all selected storms, we introduce the different analysis concepts in Sect. 4 and Sect. 5 with detailed analyses of an individual hail cell. To this end, one exemplary storm was selected from ensemble member 5, which shows a similar realization to the actual storm that moved across central Switzerland and was discussed in detail in Kopp et al. (2022) (red track in Fig. 3). This storm produced maximum hail diameters of 48.4 mm according to HAILCAST.

In this study, the storm tracks are used to enable a storm-centered view of the convective environment (Sect. 4) and initialize airflow trajectories along the tracks (Sect. 5).

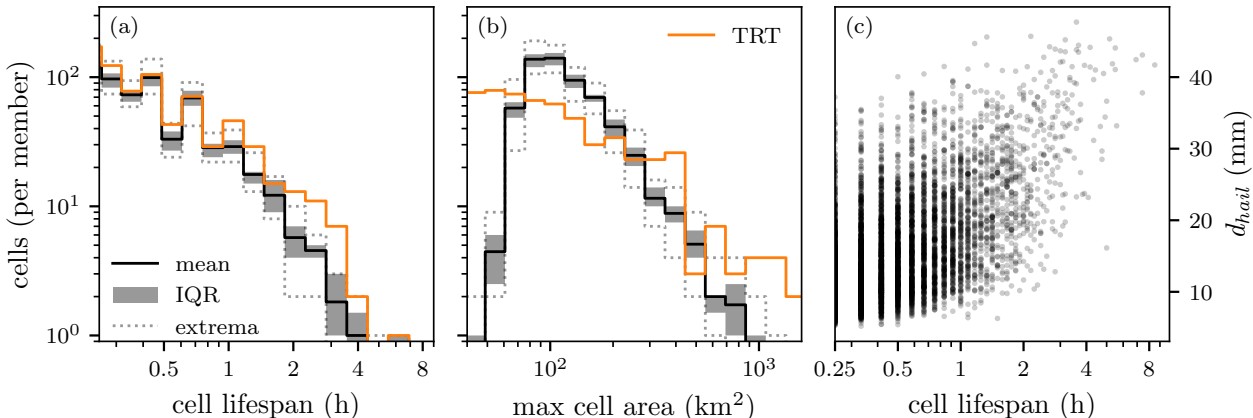

**Figure 2.** Distributions of (**a**) storm lifespan, and (**b**) maximum storm area reached throughout the life cycle, with ensemble mean (black line), interquartile ranges (IQR, gray shading), and respective extrema (gray dashed) from all members and storms of the COSMO-1E simulation. Orange lines indicate the values for the radar-observed storm tracks on the case study day (Thunderstorms Radar Tracking, TRT; Hering et al., 2004; Trefalt et al., 2023). (**c**) shows the relation between the cell lifespan and storm maximum hail diameter for all members. Note that the $y$-axis is shared between the first two panels.

## 4 Storm-centered perspective

In this section, we explore a storm-centered perspective. To this end, key environmental variables are determined along storm tracks, including various thermodynamic and dynamic parameters, such as near surface potential temperature, specific humidity, CAPE and CIN, vertical wind shear, and vertical vorticity. More specifically, a $50 \times 50$ grid-point box of all relevant variables centered at the storm location is extracted along the track of the investigated storms. Subsequently, the spatial structure of a hailstorm and its temporal development can be analyzed along its track, and atmospheric fields from multiple storms can be composited to arrive at an idealized representation of a hailstorm as simulated by the model on this particular day. Composite analysis is a widely established methodology to investigate synoptic systems (i.e., Catto et al., 2010; Binder et al., 2016) and has been recently applied to convective systems and storms (e.g., Prein et al., 2017; Oertel et al., 2020; Lin and Kumjian, 2022).

### 4.1 An illustrative example

As an illustrative example, we investigate the selected storm as described in Sect. 3.2 (red track in Fig. 3).

The storm exhibits two distinct phases, as is evident from the evolution of the vertical wind profile along the track (Fig. 4a). After initiation of deep convection at 12:00 UTC from preexisting shallow convection, a latent phase lasts for 1 h, during which maxima vertical wind velocities are restrained to $20.5\,\mathrm{m\,s^{-1}}$. Then, a first intensification (13:00–13:30 UTC) follows, when the vertical velocities increase to $31.0\,\mathrm{m\,s^{-1}}$. The start of this transition coincides well with the drop in topographical height

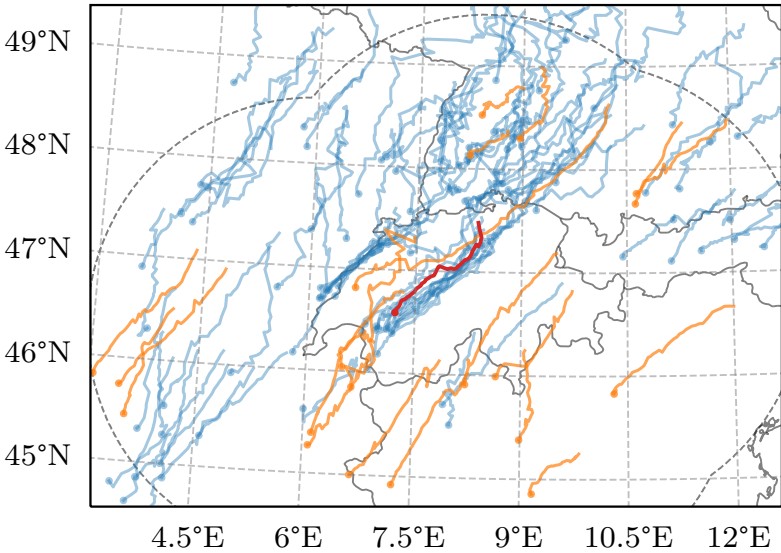

**Figure 3.** Overview of the 100 tracked storms with lifespans greater than 2.5 h that reach updraft velocities $> 25\,\mathrm{m\,s^{-1}}$ from all COSMO-1E members (blue). The start of storm tracks is marked with a dot, while the storm selected to be investigated in detail is highlighted in red. Orange tracks indicate the radar-observed tracks on the case study day with lifespans $> 2.5\,\mathrm{h}$ as classified by TRT (Hering et al., 2004; Trefalt et al., 2023). The dashed line denotes the extent of the radar domain.

from about 1300 to 800 m experienced by the storm at 13:00 UTC (Fig. 4c). Such a rapid drop in topographical height would give the storm immediate access to warmer and moister air, which in turn would fuel the storm's intensification. Preceding the most intense phase of the storm, a period of rapid intensification in updraft velocities (13:30–14:30 UTC) coincides with a clear increase of inflow mean, low level (900 hPa) $\Theta_e$ values from 341.6 K to 346.3 K (Fig. 4d). Notably, hail diameter exhibits a less rapid increase than the vertical wind (Fig. 4h). There is also a phase with lower values for $w$ and $q_c$ prior to the most intense phase, which is associated with a storm-splitting event (13:20–14:00 UTC). The maximum vertical velocity occurs at 14:40 UTC with values reaching $52.6\,\mathrm{m\,s^{-1}}$ and hail diameters of 48.4 mm (Fig. 4f). This peak in intensity is mirrored in the mid-level cloud water content (Fig. 4b). The maximum precipitation intensity produced by this storm during 5 min was $441\,\mathrm{mm\,h^{-1}}$.

The evolution of convective environment parameters is discussed in the following. Inflow[1] mean CAPE values lead the updraft maxima by an offset 0.5 h (Fig. 4e), while inflow mean 0–6 km vertical shear is initially unchanged up to 13:00 UTC and then slightly decreases after that, while 0–1 km shear evolves inversely (Fig. 4f). Further, storm maximum 0–1 km, and 0–3 km, SRH seem to mirror the evolution of the vertical wind (Fig. 4g). Lastly, the hail diameter maxima reached throughout

---

[1]The inflow region encompasses a $20 \times 20\,\mathrm{km^2}$ square box, centered 20 km ahead of the storm.

the storm lifetime coincide with the most intense phase; however, the decay in $d_{hail}$ is not as swift as the decay in intensity as given by the vertical wind (Fig. 4a). In this case, the relation between the maximum and mean values of hail diameters reported by HAILCAST within the storm environment is constant throughout the storm life.

## 4.2 Composite hailstorm structure

Next, in order to gain more statistically robust insight into the structure of the archetypal hailstorm in the simulations, composites of the selected storms were calculated. To this end, atmospheric fields from a $50 \times 50$ grid-point box centered along the storm track were collected from all storms with lifetimes $> 2.5\,\mathrm{h}$ at time steps with $w > 25\,\mathrm{m\,s^{-1}}$, in order to investigate the storm structure during the mature phase. As the storms all arise from a similar environment, they are all of similar structure, so the approach of compositing these storms is meaningful. Across the 100 storms, 2261 time steps are included in total in the composite, of which 86% feature cyclonically rotating updrafts. From the mean fields emerges the idealized structure of a storm environment. In the following, vertical cross-sections and horizontal views of various variables are presented.

First, we consider the vertical cross-section of the storm-centered composites (Fig. 5). Contours of $0.1\,\mathrm{g\,kg^{-1}}$ cloud water and ice content (chosen as the threshold for visible clouds) reveal cloud structures. A low cloud base extending down to $\sim 850\,\mathrm{hPa}$ can be distinguished in front of the rain shaft. The anvil of cloud ice precedes the slanted updraft column by more than 25 km. Composite mean updraft velocities of $16.7\,\mathrm{m\,s^{-1}}$ lead to an overshooting top protruding into the stratosphere. This overshooting top can be identified in both the mixing ratios and in the compressed isentropes (Fig. 5b). Judging by the vertical separation of isentropes near the ground, the atmosphere's stability increased after the storm passed. Interestingly, cloud water directly in the updraft column only converts to cloud ice and precipitation species as it reaches the $-38^\circ\mathrm{C}$ level, while the maxima in combined hydrometeor mixing ratio are located just below the $-38^\circ\mathrm{C}$ level. This is evident from the region of cloud ice larger than $0.1\,\mathrm{g\,kg^{-1}}$, which only marginally extends downwards from temperatures above $-38^\circ\mathrm{C}$. At temperatures above $-38^\circ\mathrm{C}$, water droplets can only freeze heterogeneously, meaning they require an ice nucleating particle to initiate freezing, while at temperatures below $-38^\circ\mathrm{C}$, water droplets will freeze homogeneously, without the aid of an ice-active particle. Whether this hints toward model limitations or is physically plausible would need to be investigated further. The microphysics parameterizations may not be ideally suited for such extreme updraft velocities, and freezing that occurs only homogeneously is an indication of this hypothesis, as heterogeneous freezing initiated by ice nucleating particles is generally considered the predominant freezing pathway (Pruppacher and Klett, 2010, and references therein). It should be mentioned here, that ice is present in the updraft in the form of graupel, even down to temperatures above $0^\circ\mathrm{C}$. However, since the cloud ice is present mostly above the level of homogenous freezing, this suggests that ice introduced by graupel below this level is not activating the freezing of cloud water droplets. Negative vertical movement only shows up in a few regions in the composite, near the ground and just behind the rain shaft. It also needs to be mentioned that the composites are centered on the positive updraft maxima, which therefore are well aligned by design, whereas the downdrafts might have slightly different locations for each storm and might cancel out when calculating the mean. Additional features that are more variable and therefore less well captured by the composite but present in individual cross-sections include detailed shelf and wall clouds extending almost down to the ground. A selected example of such features is shown in the Appendix (Fig. B1). The lack of strong, localized

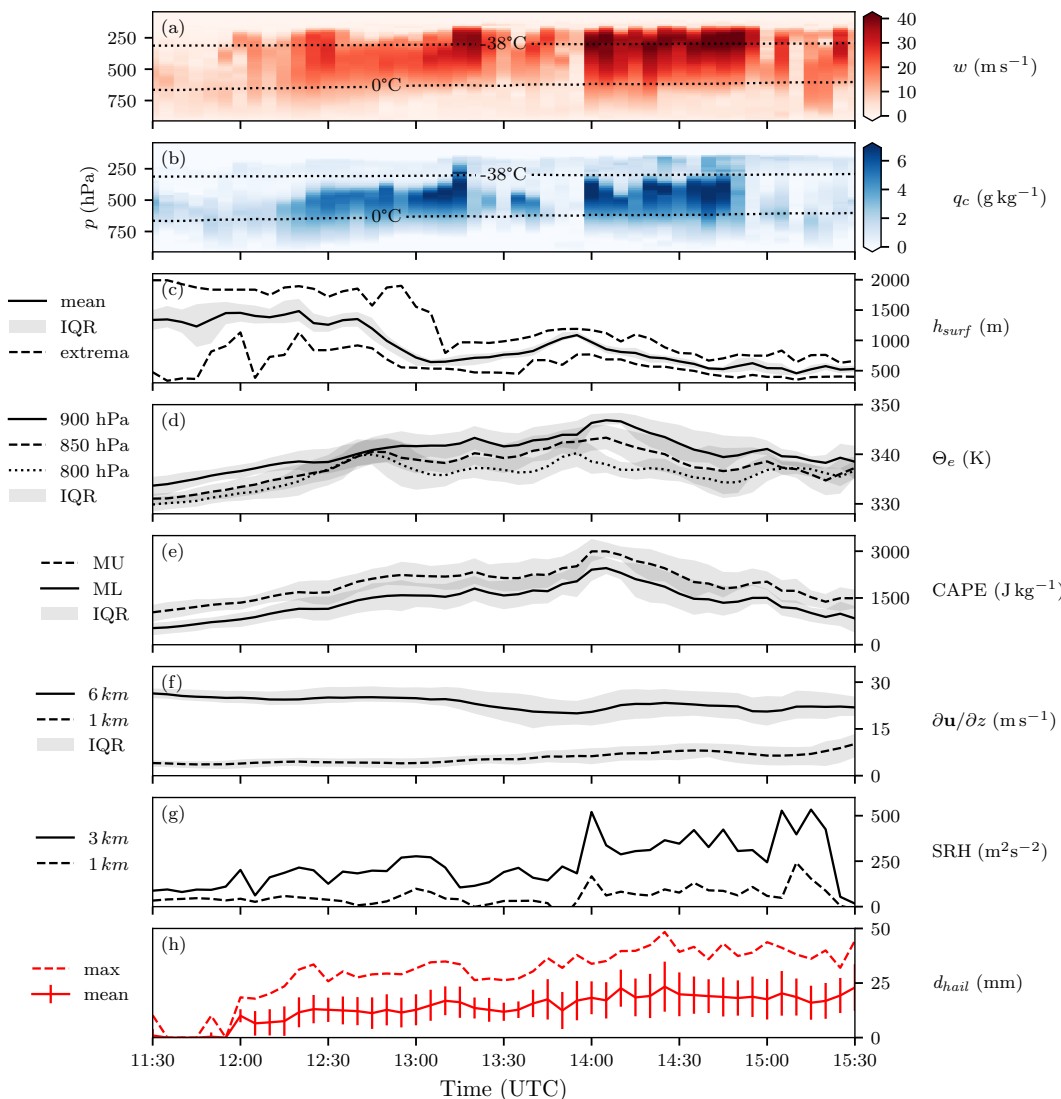

**Figure 4.** Temporal evolution of a selection of storm-centered variables (within the $50 \times 50$ grid-point box) along the red storm track in Fig. 3. The horizontal maxima of the respective variables are shown as colored pixels for the 3D variables (**a**) vertical velocity $w$, and (**b**) cloud water content $q_c$, both within the storm mask. (**c**) shows the distribution of topographical height within the storm mask (IQR as shading), (**d**) and (**e**) show the mean values of $\Theta_e$ and CAPE in the inflow region, respectively (respective IQR as shading). (**f**) shows the mean vertical 0–1 km and 0–6 km wind shear in the inflow region (respective IQR as shading). (**g**) shows the storm maximum 0–1 km and 0–3 km SRH evolution. (**h**) shows the maximum and mean hail diameter values within the storm mask as reported by HAILCAST, with the standard deviation indicated by error bars. Panels (**a**) and (**b**) also feature two dotted ambient isotherms. The inflow region encompasses a $20 \times 20 \, \text{km}^2$ square box, centered 20 km ahead of the storm. All values are from a selected storm of one ensemble member of the COSMO-1E simulation.

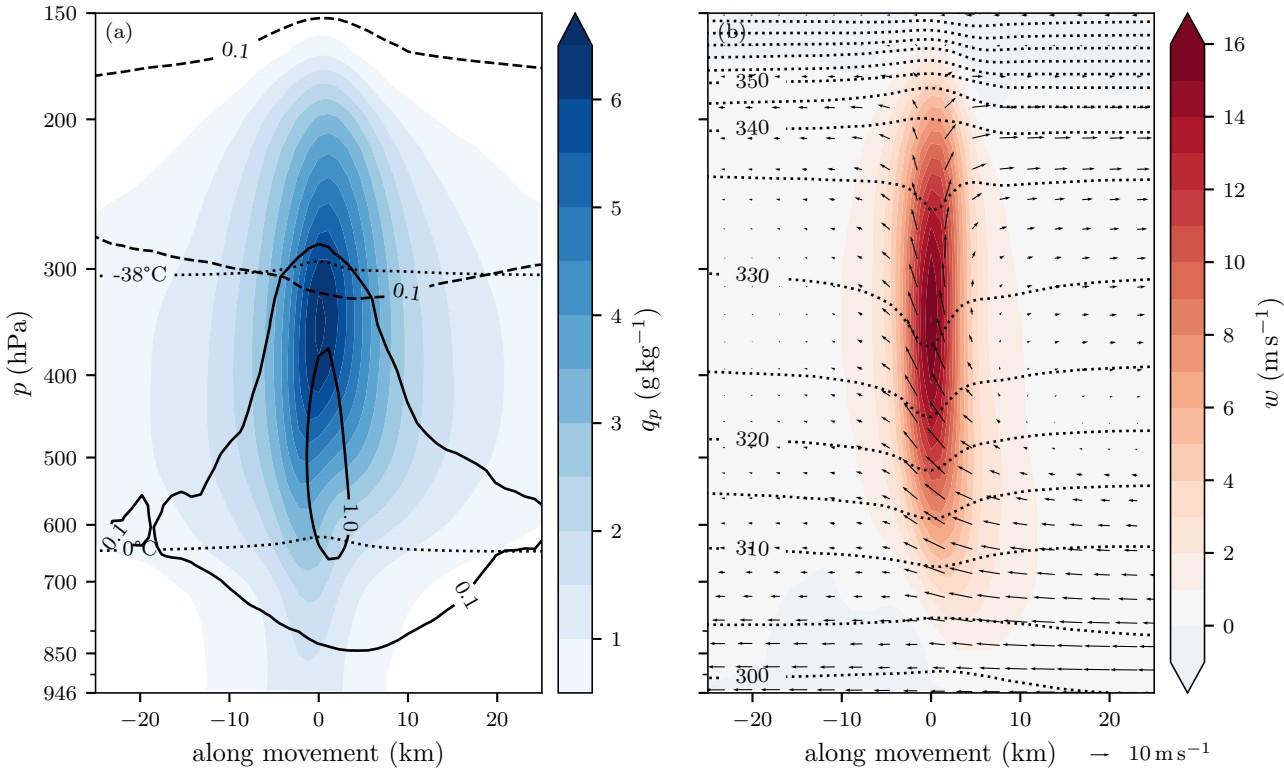

**Figure 5.** Vertical cross-section (single plane through storm center) along the propagation direction of the storm-centered composite of $n = 100$ storms with lifespans $> 2.5\,\text{h}$ at time steps with $w > 25\,\text{m}\,\text{s}^{-1}$, identified in all COSMO-1E ensemble members. The storm is moving to the right and the center (0 km) represents the storm track center. (**a**) Filled contours denote the total precipitating hydrometeor mixing ratio, solid and dashed contours represent cloud water and cloud ice mixing ratios $(\text{g}\,\text{kg}^{-1})$, respectively, while the $0°\text{C}$ and $-38°\text{C}$ levels are shown as dotted lines. (**b**) Vertical wind field (in $\text{m}\,\text{s}^{-1}$, filled contours), isentropes (in K, dotted contours), and in-plane, storm-relative wind vectors, quivers. Refer to Fig. C1 for the ensemble spread.

downdrafts points towards the possibility of large-scale subsidence as a process for balancing the storm updrafts, rather than the importance of precipitation-associated downdrafts in this specific environment. Across the composite, the standard deviation for $w$ (ensemble spread) is less than a third of the mean value. Most of the variance in the composite is located at the end of the updraft, where the overshooting top lies (Fig. C1a,b). As the storms are aligned further down, at around $400\,\text{hPa}$, differently slanted updrafts, as well as varying storm intensities would yield such a pattern.

Next, we examine the horizontal structure of the storm-centered composites (Fig. 6). The potential temperature field at the ground shows a strong gradient ($3\,\text{K}$ within $25\,\text{km}$), offset $60°$ from the storm movement direction, with a minimum located just behind the hail shaft. This minimum coincides with the pressure maximum. Both extrema are caused by the downdraft air entrained by intense precipitation and evaporative cooling forming a cold pool. The wind vector field also displays near-

ground divergence at this exact location, where the downdraft translates into the horizontal wind at the surface. 20 km ahead of the storm, an area of convergence can be observed in the 10 m wind field. The location of near-surface convergence of the horizontal wind denotes the beginning of the updraft column. Note that as the updraft core is slanted (Fig. 5b), near-surface convergence (Fig. 6a) does not overlap with the horizontal position of the updraft core aloft (Fig. 6c). Just in front of the storm center, the gust front can be determined by the near-ground composite mean wind velocity maximum of 3.3 m s$^{-1}$ (Fig. 6a). At the inflow level, which is located around 850 hPa (Sect. 5), a specific humidity maximum can be found where the air converges with a cyclonic component in the horizontal wind field. At this level, the water vapour content of the air reaches on average values of 10.6 g kg$^{-1}$ (Fig. 6b). Higher up, at 400 hPa where the updraft core is located (Fig. 5b), the near-storm horizontal winds are governed by the synoptic situation, i.e., largely determined by the pressure gradient and mostly unaffected by the presence of the storm. On average, the updraft core of the storm at this level is 9.0±4.4 km across (Fig. 6c). As the tracking of the storms is also performed at this level and on the vertical wind, the centers of the respective vertical wind maxima are well aligned and thus the resulting composite yields a very defined structure. Lagging behind the storm center by about 5 km the maximum in rain rate can be found, reaching mean values of 30 mm h$^{-1}$. The footprint of the liquid precipitation is slightly asymmetric, extending further to the left, relative to the mean storm motion. The cloud water outline, in contrast, has a slight lead on the center of the storm, which is due to the tilted updraft (Fig. 6d). Compared to the rain, the hail maxima is much more aligned with the center of the storm. Co-location of the hail and updraft maxima is to be expected, as HAILCAST does not account for horizontal advection of hailstones to other grid columns. In contrast, since graupel is explicitly included in the COSMO microphysics, it is subject to horizontal advection in the simulations. Only a small offset of the graupel maximum from the storm center exists (Fig. 6e). The location of graupel gives an upper bound on the potential advection of hail, as graupel has a smaller terminal velocity than even the smallest hailstones, giving more time for horizontal advection to take effect.

CAPE values rapidly decrease as the storm approaches. 25 km in front of the storm, CAPE values are on average 1600 J kg$^{-1}$, while they fall below 600 J kg$^{-1}$ just as the storm passes. The thunderstorm anvil cloud extends well beyond the window size of 25 km in front of the storm (Fig. 6f). Liquid precipitation following a hail event can be an important consideration for the damage caused by a storm, as rain can enter into buildings through hail-damaged skylights, windows, and roofing, causing further damage through water ingress (Ridder et al., 2020). Our analysis shows that, on average, liquid precipitation immediately followed hail fall during the passage of the storms (Fig. 6d,e).

## 4.3 Composite hailstorm life cycle

Moreover, in order to disentangle the different developmental stages of the storms, the storm-centered parameters were analyzed temporally at the stages classified as *initiation*, *mature*, and *dissipation*. To this end, storms can be "synchronized" to their respective developmental stages and directly composited, even if the storms have different lifespans and evolutionary timing. The specific timing of the developmental stages is defined as follows:

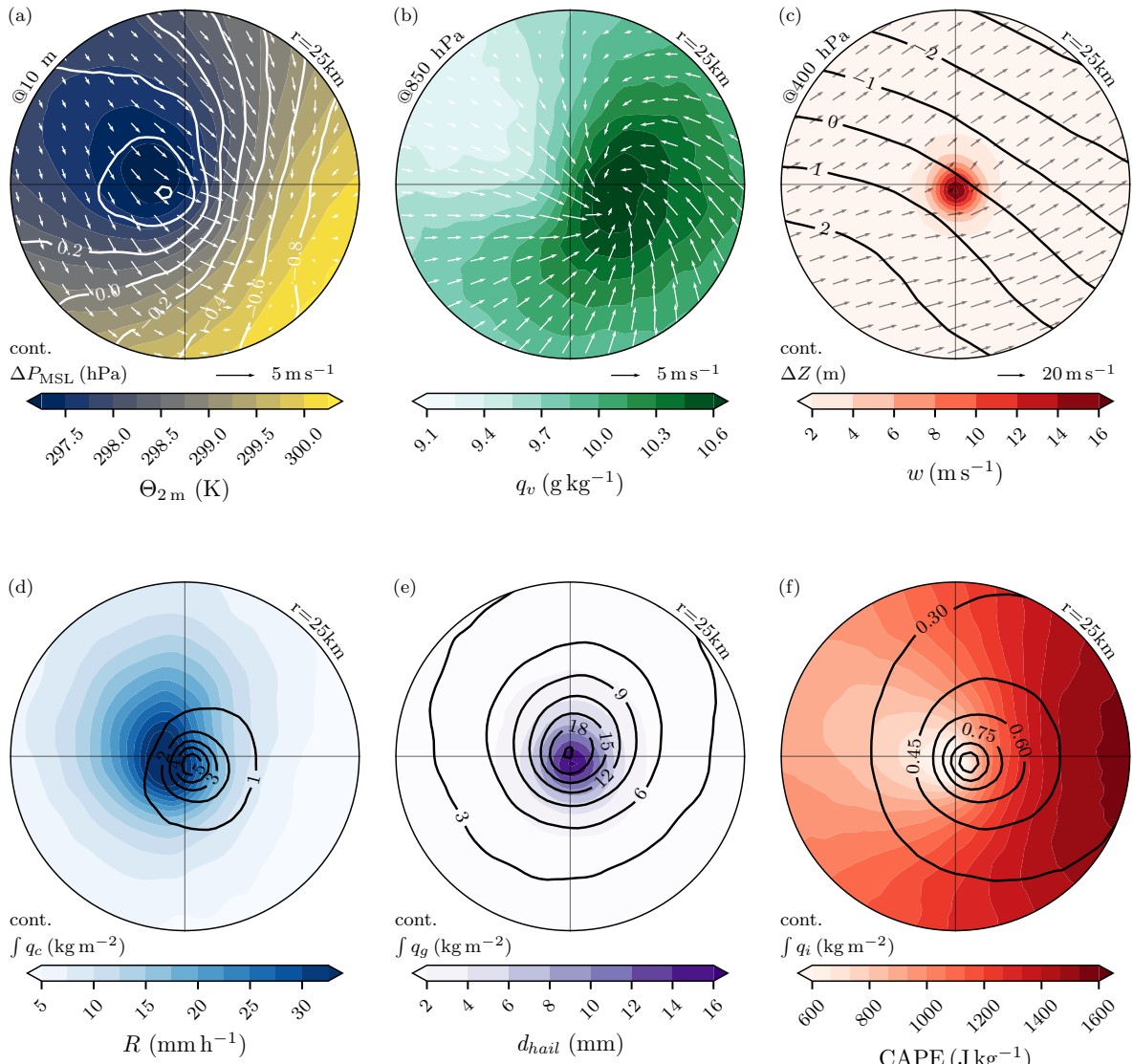

**Figure 6.** Composite analysis of the same $n = 100$ storms as shown in Fig. 5, centered on their track center (crosshair) and rotated so that their respective movement vectors align with the $x$ axis (with the storm moving to the right). The radius of the figure outline is 25 km. (**a**) Near-ground conditions: White contours show the sea-level pressure anomaly relative to the mean across the composite, and the filled contours denote potential temperature at 2 m. White arrows show the 10 m wind field. (**b**) Inflow level: Specific humidity (filled contours) and horizontal wind field (quivers) at 850 hPa. (**c**) Updraft core level: geopotential height deviation from environment mean (black contours) and horizontal wind field (quivers) at 400 hPa. (**d**) Hourly rain rate (filled contours) and column-integrated cloud water (black contours). (**e**) HAILCAST maximum hail diameter (filled contours) and column integrated graupel (black contours). (**f**) CAPE (filled contours) and column-integrated cloud ice (black contours).

- *initiation/start* — time when the storm is first detected by the tracking algorithm (more than 5 grid-points exceed $5\,\mathrm{m\,s^{-1}}$ updraft velocity).

- *mature* — moment of strongest vertical wind velocity within the storm mask.

- *dissipation/end* — time when the storm is last detected by the tracking algorithm.

From the development-stage relative view, an intuitive progression of the vertical wind arises (Fig. 7a). Prior to storm initiation, vertical wind maxima increase to $20\,\mathrm{m\,s^{-1}}$. Just after initiation, the vertical wind stagnates. As storms reach their mature phase, the vertical wind reaches a maximum, before reaching lower values again. Vertical winds exceed the baseline intensity for $\sim 1\,\mathrm{h}$. Finally, the reduced updraft intensity only becomes evident less than $0.5\,\mathrm{h}$ before dissipation. The temperature (height) of the vertical wind maximum is initially high (low) during the initiation phase, reaching a plateau shortly after that while only dipping (peaking) during maturity, and then slightly increasing (decreasing) during the last hour of the storms (Fig. 7b). The hail diameter generally follows the evolution of vertical wind closely, while exhibiting a delay during the initiation and a less pronounced, broader maxima during the mature phase (Fig. 7c). CAPE decreases steadily during all developmental phases while reaching its minimum during the dissipation (Fig. 7d). There is no increase in environmental CAPE, as the storm reaches its mature phase. It should be noted here, that neighbouring storms might influence/reduce CAPE in the inflow box. The storm area follows the same general shape as the updraft, although it exhibits a minor delay of 15 min during the mature phase (Fig. 7e). Inflow air initially flows over higher terrain during the initiation phase and passes over lower terrain as it feeds the dissipating storm (Fig. 7f). Contrary to the sequence of topographical height and storm vigor seen in 4, no such evidence was found when investigating all storms in this regard. During the dissipation phase, starting $\sim 0.5\,\mathrm{h}$ prior to the storm's end of life, the mean inflow altitude[2] markedly separates from the topography, while the inflow is on average around $600\,\mathrm{m}$ AGL during the storm's lifetime, and this changes to $800\,\mathrm{m}$ AGL just before dissipation (Fig. 7g, see also Sect. 5). Throughout the storm's lifetime, the bulk of the inflow originates from $330$–$900\,\mathrm{m}$ AGL. Previous studies found the inflow level height of simulated supercells to be between $1400$ and $1800\,\mathrm{m}$ AGL (Thompson et al., 2007; Nowotarski et al., 2020), while in idealized simulations of supercell low-level mesocyclones, Coffer et al. (2023) found the inflow to be around $200$–$400\,\mathrm{m}$ AGL. The number of inflow trajectories filtered for ascent strongly increases up to a maximum just before the storms reach their mature phase (Fig. 7h; see later Sect. 5).

### 4.4  Storm-environmental parameters

Finally, a comprehensive view of the joint distribution of storm-environmental parameters is facilitated by investigating values within a $50 \times 50$ grid-point box centered on the storm track for the same selection of storms as in Sect. 4.3. It allows us to analyze the relationships and correlations between different variables. One notable correlation is between the hail diameter ($d_{hail}$) and the vertical wind velocity ($w$), with a correlation coefficient of 0.776 (Fig. 8a). This strong positive correlation suggests that larger hail diameters are associated with higher vertical wind velocities. In our analysis, CAPE does not correlate with hail

---

[2]Mean height above ground level of the inflow trajectories in the period $-60$ to $-30$ min before reaching the storm.

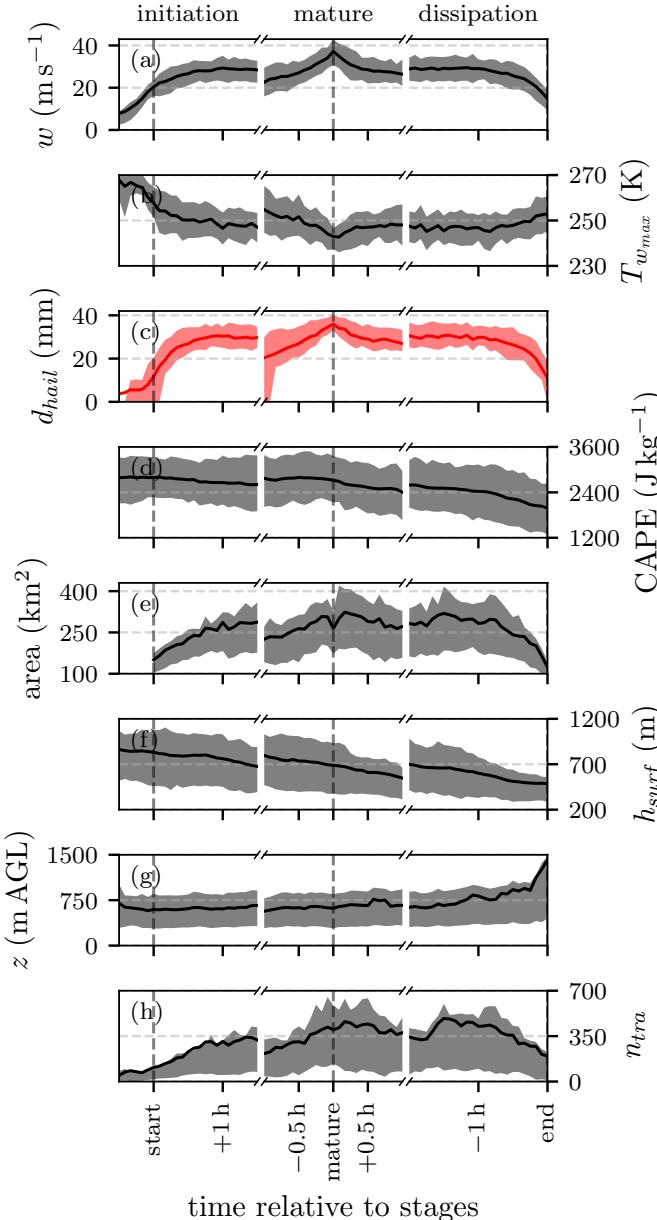

**Figure 7.** Values for select variables within a $50 \times 50$ grid-point box centered on the storm position during three developmental stages of $n = 100$ storms identified in all COSMO-1E members with lifespans $> 2.5\,\text{h}$ that reach updraft velocities $> 25\,\text{m s}^{-1}$. Shown are (**a**) vertical wind maximum, (**b**) ambient temperature at the height of vertical wind maximum, (**c**) maximum hail diameter $d_{hail}$, (**d**) maximum CAPE, (**e**) storm area, (**f**) mean surface elevation, (**g**) mean height above ground level of the inflow trajectories in the period $-60$ to $-30\,\text{min}$ before reaching the storm, and (**h**) number of filtered trajectories. $w$ and $d_{hail}$ are masked by the storm track mask. The solid line represents the mean value across all storms, while the shaded area shows the interquartile range.

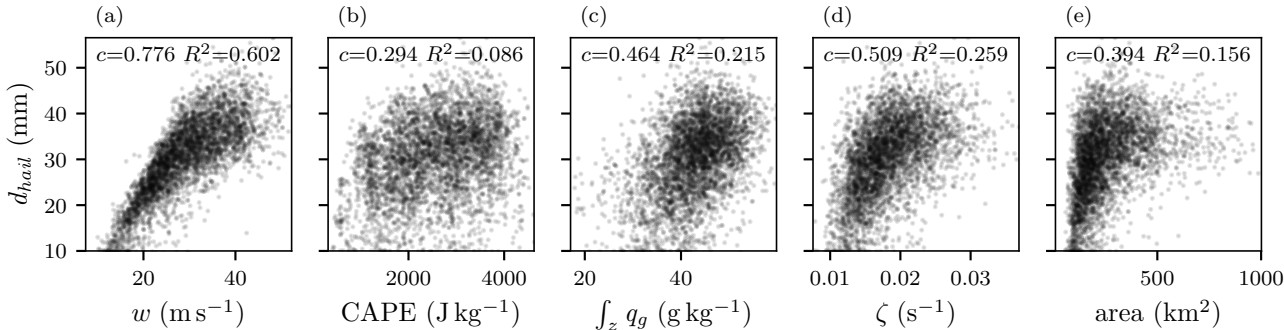

**Figure 8.** Scatter matrix with per time step storm-environmental parameters (considered in box with $50 \times 50$ grid-points) from $n = 100$ storms with lifespans $> 2.5$ h that reach updraft velocities $> 25 \, \text{m s}^{-1}$. All values but the area are the per-time step maxima found within the environmental box. Correlation and $R^2$ values are indicated in each permutation. COSMO-1E simulation, all 11 members.

diameter, nor with $w$ (Fig. 8b). On thermodynamic grounds, $w$ would be expected to scale with $\sqrt{2\text{CAPE}}$. A plausible explanation is that local updraft accelerations are not solely determined by buoyant forcing. Vertical pressure-gradient accelerations, which can arise independently of CAPE (Markowski and Richardson, 2010), may enhance or diminish the buoyancy-induced updraft, while precipitation loading can further suppress vertical velocity. Consequently, these dynamical factors can obscure any direct relationship between $w$ and CAPE within individual storms. Further, the vertically integrated graupel shows a weak correlation coefficient with the hail diameter of $0.464$ (Fig. 8c). One approach to estimating hail is from the integrated graupel and in our case this approach would not explain the whole range of hail diameters reported by HAILCAST. Larger hail diameters are also found at time steps when the vertical vorticity of the storm was highest ($c = 0.509$, Fig. 8d). Generally, large footprints are also associated with larger maximum hail diameters, however, large hail diameters can also be found in storms with smaller footprints ($c = 0.394$, Fig. 8e). Finally, it is important to note that this analysis only provides a statistical view of the storm environment. To obtain a more physically meaningful perspective of the air feeding the storm, we turn to inflow trajectories, which will be discussed in the following section. It should be noted here, that due to the one-dimensional nature of HAILCAST, the effect of increased updraft width — which has been established as an important factor for hail growth (e.g., Nelson, 1983) — is not entirely accounted for. Thus, any correlation between environmental parameters and HAILCAST hail size would not include the effect of the updraft width.

## 5 Lagrangian analysis: inflow trajectories

In this section, we discuss the Lagrangian perspective of the air parcels processed by the storm. Airflow trajectories were calculated using LAGRANTO (Wernli and Davies, 1997; Sprenger and Wernli, 2015) and 5-min wind fields from COSMO-1E. At each time step when a storm was active, points within the storm mask, laying on a $275 \times 275 \, \text{m}^2$ horizontal grid, at $z = 5000$ m above mean sea level were selected as trajectory starting points. These accommodations were made to ensure the

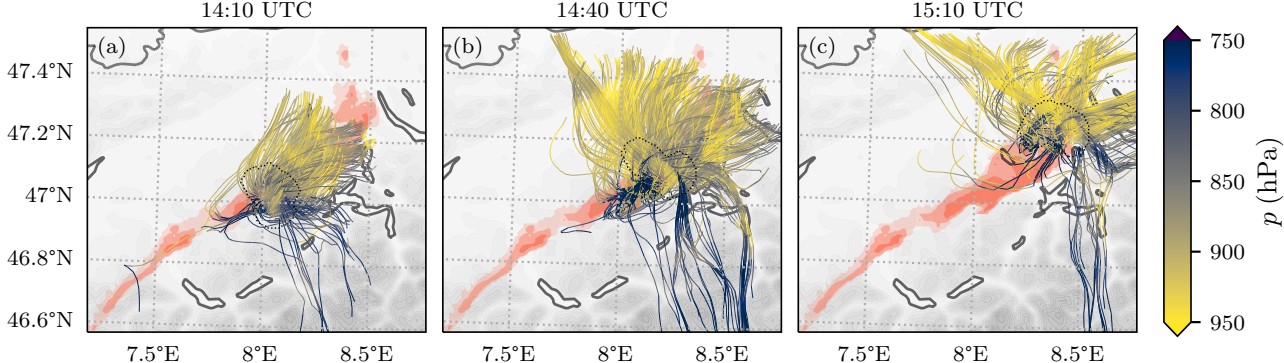

**Figure 9.** 2 h backward trajectories that are feeding a selected hailstorm updraft on 28 June 2021 during its most intense phase at 14:35 UTC (**b**) and 30 min prior and after the most intense phase (**a** & **c** respectively). Only trajectories that experience a change in pressure of > 600 hPa are shown. The trajectories are colored with their local respective pressure. The red-filled contours show the maximum updraft swath areas that exceed velocities of 10, 20, and 30 m s$^{-1}$ respectively, while the black dotted contour encloses the current storm mask. The gray shading represents topography height, while grey lines denote the national borders and lakes. COSMO-1E simulation, one member. An animated version of this figure is included in the supplement.

robustness of the results. By starting the trajectories on a fine grid and in high-frequency time steps (5 min) we essentially produce an ensemble of trajectories, which reflects the inherent uncertainties in the Lagrangian methodology. 2 h backwards as well as forwards trajectories were calculated with LAGRANTO using a 1 min time step, while the trace variables were recorded every 5 min, consistent with the available model output time step. The resulting trajectories were filtered for ascent with the criterion of ascending at least 600 hPa within the 2 h centered on the time of storm intercept (when the trajectories traverse the 5000 m level), similar to filtering criteria used in warm conveyor belt studies (i.e., Heitmann et al., 2024). This filtering criterion applies to 10.4% of the 13 million trajectories.

To highlight the path of the inflow, we consider the trajectories arriving in the selected storm's updraft (ensemble member 5, as explained in Sect. 3.2) at the time of highest intensity (14:35 UTC). While the storm is propagating in a north-easterly direction, the bulk of the inflow is moving in the opposite direction. While the main inflow is initially broad and parallel, it narrows down and converges as it approaches the storm all while remaining close to the ground until arrival at the updraft (Fig. 9). Surprisingly, some stray parcels are also advected across the main Alpine crest before entering the updraft while still rising more than 600 hPa within the storm. Further, a broadening of the inflow sector can be observed at later times. While the inflow trajectories are almost parallel at 14:05, they markedly broaden as the storm passes maturity and approaches its dissipation.

Next, we consider various microphysical and thermodynamic properties of the air being processed by the storms. To this end, we align all of the inflow trajectories and their respective trace variables according to their intersection time with the storm (Fig. 10). The majority of the ascent of air parcels in the storm updraft is very rapid. The parcels are lofted from near-

ground level to the tropopause in mere tens of minutes. During the inflow period, up to $-0.5\,$h, the CAPE at the parcel height increases with a rate of $0.21\,\mathrm{W\,kg^{-1}}$, while the CIN is reduced at a rate of $-0.0080\,\mathrm{W\,kg^{-1}}$ in the same time. CAPE is rapidly consumed, as the parcel ascends. The main effect of the decrease in the final minutes before entering the storm is that the parcels are already vertically displaced, so the portion of CAPE below the parcel is not accessible anymore. The spread in parcel pressure during the inflow phase is very small, while there is a larger spread in the anvil outflow phase (Fig. 10a).

To disentangle interactions between the different precipitation species and their effect on the inflow parcels, changes in $\Theta_e$ are investigated. The evolution of $\Theta_e$ during the inflow phase closely follows that of CAPE. After intercept, $\Theta_e$ returns to the same values as were present during the inflow phase. The parcels exclusively approach saturation during the intercept period, while moistening starts to occur at $t = -1\,$h relative to intercept (Fig. 10b). The development of the different microphysical species along the trajectories is as follows: The first species to increase in concentration is rain, followed by cloud, graupel, ice, and finally snow (Fig. 10c). Although some of the intercept-relative variables are more easily explained, such as the values in Fig. 10a, others have less trivial explanations. $\Theta_e$ is conserved through adiabatic processes, as well as during condensation of the parcel moisture. In the absence of moist convection, near-surface $\Theta_e$ is expected to increase during the day, due to surface radiative processes and surface moisture sources. This effect can be observed far upstream of the storm (Fig. 10b, $-2$ to $-1\,$h). As latent heat is released in the updraft, $\Theta_e$ is expected to remain constant. However, a clear decrease in $\Theta_e$ with a minimum at the time of intercept was found (Fig. 10b). Several, $\Theta_e$-nonconserving phenomena offer explanations for the decrease and increase seen in $\Theta_e$ during inflow and outflow phases, respectively. Sensible heat flux from the rain entering the air parcel from above would reduce $\Theta_e$, as the rainwater is colder than the air it enters and thus extracts heat energy from the parcel. As the air parcel is sub-saturated during the inflow period, there is also evaporative cooling from the infalling rain, however, this would not affect $\Theta_e$, as the moistening of the parcel cancels it out[3]. Most of the inflow occurs in the warm phase, however, with the melting level at $\sim 650\,$hPa there are also some mixed-phase processes to consider. Graupel falling into the inflow parcel would also reduce $\Theta_e$ of the parcel, through the same effect discussed for the rainfall, with added cooling through the latent heat needed to melt the solid phase. After the storm intercept in the mixed-phase cloud, the liquid species freeze to cloud ice, snow, and graupel, releasing latent heat, and thus increasing $\Theta_e$. Vapour deposition of the gaseous phase to the ice phases is also expected to increase $\Theta_e$.

Another curious observation is that rain precedes the presence of cloud water in the inflow parcels (Fig. 10c). This rain can not be produced by the parcel itself, as the rain is present in sub-saturated conditions, and at times when no cloud water is present. As such, it must be rain that is falling into the updraft parcels from higher levels. Other studies have found fewer interactions of hydrometeors with the inflowing air (i.e., Coffer et al., 2023). These findings form the basis for the summary and conclusions in the next section. There might be some limitations associated with this finding, connected to the

---

[3]As the trajectories are not a closed system, there might be influence from, e.g., sensible heat transfer from colder rain falling into the warmer air beneath and removing heat energy, effectively reducing $\Theta_e$ in the parcel. However, this effect is expected to be minor, as the temperature difference between the parcel and the infalling rain is limited.

convection-permitting, and not convection-resolving nature of the simulations, and small-scale features not being properly re-solved. However, it is evident that the $Q_r$ is the first species to increase in concentration, even at $-0.5\,\mathrm{h}$ away from reaching the storm, which would account for several grid-points in front of the storm (Fig. 10).

## 6 Summary and conclusions

This study presents a detailed analysis of severe hailstorms that occurred in Switzerland on 28 June 2021 using the high-resolution convection-permitting ensemble hindcast COSMO-1E and a combination of Eulerian and Lagrangian diagnostics. Our comprehensive approach combines object-based tracking techniques for hailstorms, Eulerian analysis of storm-associated atmospheric variables, and Lagrangian analysis of air parcels feeding into a hailstorm's updraft. Through a novel implemen-tation of object-based tracking, we established the storm characteristics and compared them with recorded radar observations, qualitatively validating the simulated storm track against reality. We then systematically analyzed storms that occurred in any of the simulated ensemble members with a lifespan greater than 2.5 h and updraft velocities exceeding $25\,\mathrm{m\,s^{-1}}$, which revealed several long-lived and intense hailstorms within the simulation domain, mainly concentrated to the north of the Alpine crest. COSMO-1E with HAILCAST simulates, for this case, realistic hail tracks in terms of storm lifetimes, storm area, propagation velocity, and direction (Fig. 2 and 3).

The Eulerian perspective, focusing on atmospheric parameters and fields around the storms, allowed us to decompose the life cycle of hailstorms into initiation, mature, and dissipative stages. We investigated the spatial structure of the composite of 100 simulated storms that resemble the observed hailstorm, which creates an idealized picture of their vertical profile and horizontal environment. The latter revealed that the simulated hailstorms propagate in the direction of gradients of CAPE and low-level specific humidity. They also confirmed that rain-after-hail was prevalent for the storms occurring on this day (Fig. 6).

Lagrangian analysis provided physical insights into the inflow dynamics and source regions of air ingested by the hailstorm. We confirmed that most inflow air was channeled near the surface and could trace the evolution of relevant variables along the trajectories, including the reduction of CAPE and CIN, and the progression of various hydrometeor species from liquid to frozen states. The presence of rain prior to parcel saturation and cloud water formation, and the changes in equivalent potential temperature suggest complex interactions between the precipitation species falling into the storm's updraft, thus transporting heat vertically. The analysis of the airflow into hailstorms and investigation of the Lagrangian evolution of CAPE, CIN, and hydrometeors leads to the following conclusions: (i) there is a rapid decrease of CAPE and CIN from $1100\,\mathrm{J\,kg^{-1}}$ and $-60\,\mathrm{J\,kg^{-1}}$ when air parcels converge into the updraft region; (ii) the hydrometeor sequence shows rain predates all other cloud and hydrometeor species in the inflow air parcels; (iii) the origin of hailstorm air is from various regions, notably including air being drawn into storms and feeding the updraft from across the Alpine crest (Fig. 7, 9, and 10). Our analysis provides insights into the temporal evolution of the storms and expands on the tracer analysis in the steady-state composite from idealized simulations performed in Lin and Kumjian (2022). We were able to show that the inflow airstream essentially detaches from the lowest 800 m up to 30 min prior to dissipation of the storms (Fig. 7g).

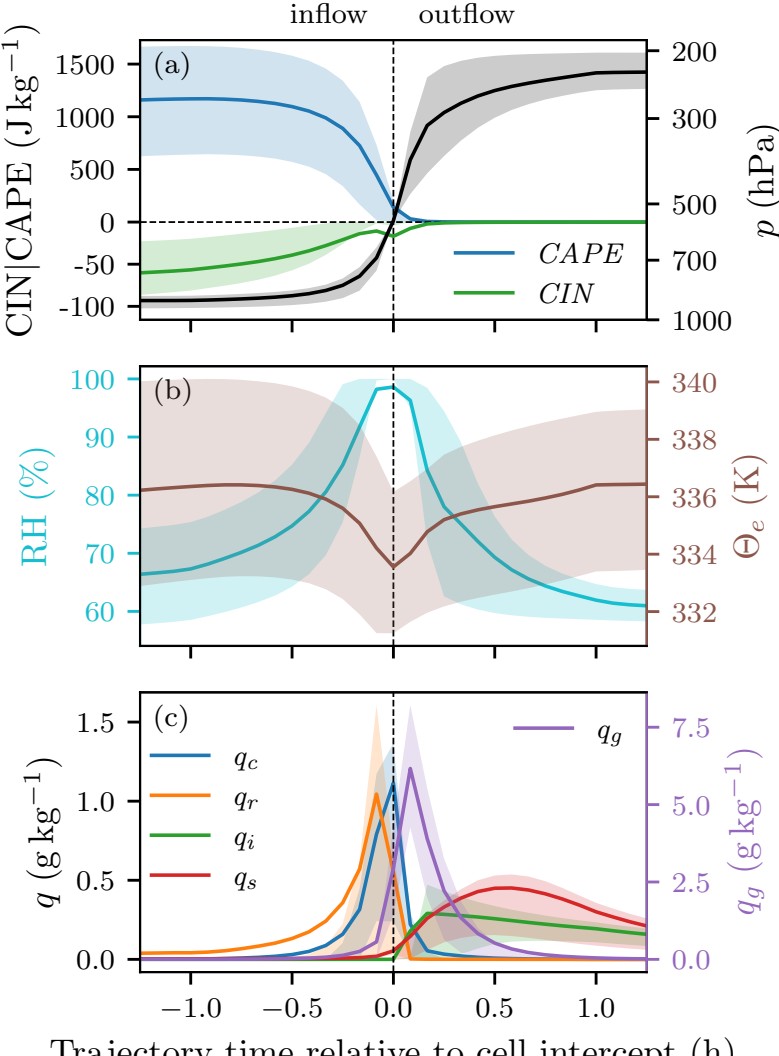

**Figure 10.** Various trace variables from all time steps of all $n = 100$ storms with lifespans $> 2.5\,\mathrm{h}$ that reach updraft velocities $> 25\,\mathrm{m\,s^{-1}}$ as a function of trajectory time relative to intercept of the trajectory with the storm at $z = 5000\,\mathrm{m}$ ($0\,\mathrm{h}$ on the $x$-axis). The solid lines show the mean for all trajectories of all storms and time steps at a given trajectory time, while the shaded areas indicate the interquartile range. The dotted line represents the storm intercept time. (**a**) CAPE & CIN at parcel height on the left axis and parcel pressure on the right axis. Note that CIN is on a scale different from CAPE. (**b**) Relative humidity and $\Theta_e$ of the parcel. (**c**) Microphysical species (cloud, rain, ice, snow, and graupel) with $q_g$ on the right axis, with different scaling. COSMO-1E simulation, all 11 members.

This study highlights the strength of combined Lagrangian and Eulerian analysis methods in furthering our understanding of the complex processes governing hailstorm evolution. The demonstrated object-based and Lagrangian approach is a substantial step forward in severe storm studies, offering deeper insights than typically provided by Eulerian analysis alone. The

storm-centered perspective proved useful in analyzing the storm's structure and evolution, while the inflow trajectories provided insights into the pathway of the air feeding the storm. The storm-centered composites could also prove to be useful in analyzing the typical storm structure as simulated by convection-permitting climate simulations, whereas the available output frequency and vertical resolution available from climate simulations would likely not suffice for the Lagrangian analysis. The Lagrangian methodology could also be employed to diagnose the moisture sources of the hailstorms, or investigate the evolution of vorticity in the air processed by the storm, and how this influences the rotation of the storm core. Future work could also look into the broadening of the inflow as a storm moves towards dissipation mentioned in Sect. 5. In conclusion, the ensemble simulation-based perspective leveraged in this study enhances our ability to explore hailstorm initiation, intensification, and dissipation mechanisms in a robust way. It underscores the importance of considering both storm-scale dynamics and mesoscale environmental conditions in predicting and interpreting hailstorms. This research framework can be adapted and extended to other severe weather phenomena, thereby helping in developing better adaptive strategies for managing and mitigating the risks associated with such events.

We would like to close this study by briefly reflecting on the more general implications of this model-based study for hail research, and its main limitations. For many decades, research on the dynamics of hailstorms was mainly driven by radar meteorology — for mainly three reasons: (i) radar data provide the most complete and detailed picture of the three-dimensional structure of hailstorms and their temporal evolution, (ii) the formation of hail is complicated and not routinely implemented in operational weather prediction models, and (iii) kilometer-scale resolution (at least) and the explicit treatment of deep convection is required to realistically simulate the evolution of thunderstorms associated with hail. As demonstrated in this study, the advent of high-resolution weather prediction ensembles with a hail diagnostic enables detailed studies of hailstorm formation, their evolution, and in particular their interaction with the environment. Clearly, such a novel approach also comes with limitations. At the moment, it appears to be very difficult to validate some of our findings with observations, because of limitations of the observational network to cover the complex 3-dimensional flow evolution near hailstorms. Further, the convection-permitting, rather than convection-resolving nature of the simulations used in this study impose limitations on the storm structures that are represented (i.e., Bryan et al., 2003; Jeevanjee, 2017). Also, some results presented in this study might be very specific to the case investigated (i.e., not representative of hailstorms in Switzerland in general) and they might be model-dependent. In addition, the research results gleaned in this study are subject to biases inherent in the hail diagnostic. We, therefore, suggest that similar investigations will be done for other cases and other models, in particular also with modeling systems where hail is not diagnosed in the vertical column but explicitly simulated by the microphysics parameterization.

*Code and data availability.* The tracking algorithm used in this study (Appendix A) is available under https://doi.org/10.5281/zenodo. 12685276. Storm track data, storm-centered composite, and Lagrangian trace variables are available through the ETH research collection https://doi.org/10.3929/ethz-b-000702890. The full simulation output is available from the corresponding author upon request.

### Appendix A: Tracking algorithm

In this section, we describe the tracking algorithm `cell_tracker` in detail. The full Python code for the tracking algorithm is available at https://git.iac.ethz.ch/scclim/cell_tracker. Some of the parameters and thresholds chosen for this study were described in Sect. 2.2 and other default thresholds will be given throughout the following description, wherever a new algorithm parameter is introduced. The tracking algorithm was designed to be variable agnostic, and should, after tuning the thresholds and parameters, work with any two-dimensional field. To keep this description as universal as possible, wherever a threshold or parameter refers to the specific intensity value of the input field, we use intensity units (`iu`). The tracking functionality is provided by the function `track_cells` in the repository.

There are several approaches to tracking atmospheric objects such as convective storms and other atmospheric phenomena described in the literature (e.g., Neu et al., 2013; Gropp and Davenport, 2021; Meredith et al., 2023; Liu et al., 2024). Recent studies (Schär et al., 2020; Rüdisühli, 2018; Rüdisühli et al., 2020; Schemm et al., 2020) described an on-the-fly feature tracking algorithm, based on overlapping areas of the features identified at the previous and current time steps. Importantly, as such algorithms run online (that is, during model run-time), they can benefit from a very high temporal data resolution, and simply using overlapping areas is enough to track features. For data with less temporal resolution, a simple overlap association is not sufficient for small, fast-moving objects. This shortfall can be combated by implementing dynamical tracking, where the search area at a given time step is not just taken from the location of the tracked feature at the previous time step, but informed by previous feature movement. An approach to using horizontal wind fields at multiple model levels to guide the search area in the next time step was implemented in Purr et al. (2019), allowing tracking in lower time resolutions. However, their algorithm does not account for the splitting and merging of storms and is reliant on wind fields being available during tracking. TRT (Thunderstorms Radar Tracking; Hering et al., 2004; Trefalt et al., 2023)), a radar-based tool for thunderstorms nowcasting that is used operationally by MeteoSwiss, is another tracking algorithm that does not account for splitting and merging but has an adaptive threshold implementation. Because it benefits from the high temporal resolution of 2.5 min for the radar volume scans, it does not require dynamical tracking. TRT has been further developed (Thunderstorm Detection and Tracking (T-DaTing); Feldmann et al., 2021), adding simple two-way splitting & merging support and optical flow to predict storm movement.

### A1 Identification and segmentation

First, features are identified in a two-dimensional field using thresholds for various parameters described below. A feature constitutes a set of grid points (`gp`) and has certain properties such as size, center of mass, magnitude, etc. Local maxima $M$ are then defined as connected sets (4-connectivity) of grid points with magnitudes strictly greater than the magnitudes of all pixels in the direct neighborhood of the set. Local maxima must fulfill the minimum distance threshold `min_distance = 6` and will be neglected otherwise. The `prominence` of a local maxima $M$ is defined as the magnitude difference between $M$ and the lowest isopleth encircling only $M$ and local maxima with magnitudes smaller than $M$. Local maxima must satisfy the minimum prominence threshold of `prominence = 10iu` and will be neglected otherwise (function `label_local_maximas`).

From a synthetic input field (Fig. A1a) with 6 objects in total, the segmentation algorithm classifies and labels 3 objects that fulfill the default tracking parameters (Fig. A1b). As visualized in Fig. A1c, initial object 1 does not fulfill the `threshold` criteria, while objects 2 through 6 exceed the `threshold`. Objects 2 through 4 are encircled by a contour greater than `threshold`, but object 4 is segmented due to sufficient `prominence`, objects 2 and 3 do not fulfill this criterion. Object 5 is segmented by the simple `threshold` criterion, while object 6 is discarded as it does not satisfy the `min_area` filter. Using local maxima obtained through the previous steps as seeds, a watershed segmentation algorithm is applied (Najman and Schmitt, 1994; van der Walt et al., 2014). The watershed algorithm treats magnitudes as topography (elevation) and floods basins from the seeds until basins attributed to different seeds meet on watershed lines (Fig. A1, objects 2, 3 and 4). These basins are then associated with a unique label whose area encompasses a feature. The basins extend from $M$ until the intensity drops below the threshold `threshold = 5iu` (function `watershed`). In order to extend the label area spatially and increase tracking robustness, a binary dilation is applied through kernel convolution, which expands the label area into background regions by `aura = 3gp` while avoiding overlaps with neighboring label areas (function `expand_labels`. The resulting feature area must be larger than `min_area = 16gp`.

## A2 Forward movement anticipation

Using a geometrically decaying weighted mean of the feature propagation vector history of the last `dynamic_tracking = 4` time steps, the labeled area from the previous time step is shifted towards the expected position of the feature in the current time step and is used as a search mask (function `advect_array`). If the feature first emerges and has no previous vector, a flow field is extracted from nearby features and used as an initial movement vector (function `generate_flow_field`). A limit is imposed on the maximum value for the advection of the search mask (`v_limit = 10gp`) to avoid erratic behavior and unphysical representation in the tracks of the atmospheric objects (e.g., track skipping along squall line).

## A3 Correspondence

Finding corresponding features between the current and previous time steps is implemented by computing a tracking probability score for all correspondence candidates. First, for any given feature, active in the last time step, correspondence candidates in the current time step are determined based on the nonzero overlap criterion from the advected search mask (function `find_correspondences`). Next, all correspondence candidates are clustered into groups whose correspondence assignment can be solved independently (function `find_cluster_members`). If multiple corresponding features are found within an independent cluster, all possible candidate combination permutations are analyzed holistically for their overall tracking probability score within that cluster (function `correspond_cluster`). A weighted combination of the overlap surface area ratio and the feature size ratio is taken into account to calculate the tracking score (function `calculate_score`). By default, the overlap and surface area ratio are equally weighted (`alpha = 0.5`) to construct the score. A similar solution to the correspondence problem was implemented by Rüdisühli (2018) and the probability score was directly adapted. Choosing the most likely correspondence combination with the highest probability score leads to the attribution of features that either start existing, carry on, cease to exist, split into multiple features, or are merged into another feature. The storm exhibiting the

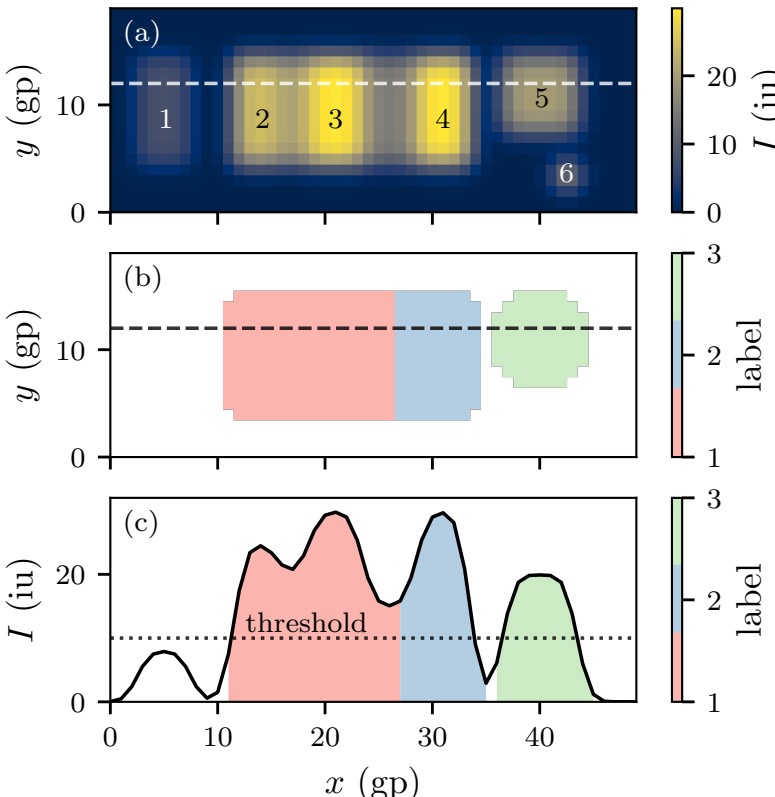

**Figure A1.** (a) Synthetic intensity field featuring 6 objects with varying intensities and areas. The dashed line indicates the position of the cross-section in panel **c**. (b) Object labels extracted from the intensity field in panel **b**. The dashed line indicates the position of the cross-section in panel **c**. (c) Intensity along the cross-section indicated in panels **a** & **b** with the labeled objects indicated in color.

largest area keeps the ID (parent), while the other is assigned to a new ID (child) while retaining the parent-child relationship. During merging events, the more long-lived feature retains its ID, while the other merge participant ceases to exist, but while retaining the merge relationship to the merge target.

Clusters with up to `cluster_size_limit = 16` correspondence possibilities resulting in a permutation of size $2^{16}$ can be scored efficiently (in $< 1\,$s, single core, Intel® Xeon® CPU E5-2690 v4 @ 2.60 GHz). If the cluster size exceeds `cluster_size_limit`, the candidates with the smallest overlap are pruned if they can be assigned to other features, so only the 16 candidates with the largest overlap area are considered for the correspondence (function `prune_cluster`). Exceptions where cluster pruning is not sufficient to reduce the cluster size, further reduction in cluster size is handled by `crude_correspondence`. The crude assignment starts by assigning the object with the smallest area to the candidate with the best score until the `cluster_size_limit` is respected, after which the standard correspondence resumes. Note that for the crudely assigned correspondences only continued survival and no splitting/merging is possible. This significantly im-

proves tracking performance for large clusters. However, it should be mentioned here that reaching large cluster sizes can hint towards non-optimal choices for tracking parameters and fine-tuning thereof should be strongly considered, rather than relying on cluster-pruning and crude correspondence.

## A4 Swath gap filling

One application of the storm tracks includes reducing the 'fishbone effect', a term coined in Lukach et al. (2017), describing the discontinuous hail swaths caused by the low temporal sampling of fast-moving and short (in the direction of movement) storms. The 'fishbone effect' could especially bias damage models, as the hail-affected area is underestimated, especially for large hail diameters. Gaps in the hail swaths are present, even with a relatively high temporal sampling of 5 min (Fig. A2a).

To fill the hail swath gaps (function `fill_gaps`), storm footprints from two adjacent time steps can be linearly interpolated
to form an intermittent storm footprint at a virtual time step $\mu$. The intermittent storm footprints are then translated to their linearly interpolated positions determined from the storm movement vector $\boldsymbol{v}$ and compounded using:

$$\Psi^n = \max\left(\left[\frac{\mu}{\kappa}\psi^n_{i+\boldsymbol{v}\frac{\mu}{\kappa}} + \left(1 - \frac{\mu}{\kappa}\right)\psi^{n+1}_{i-\boldsymbol{v}\left(1-\frac{\mu}{\kappa}\right)}\right]_{\mu\in\{\mathbb{N}\leq\kappa\}}\right), \tag{A1}$$

where $\Psi^n$ is the linearly interpolated swath at time step $n$, $\psi^n_i$ is the storm footprint at time $n$ and position $i$ as determined by the tracking algorithm, $\kappa = \Delta t_{virt.}/\Delta t$ is the number of virtual time steps per simulation output time step. Translations are
590 performed at the grid point level, and the derivations of $\boldsymbol{v}$ are rounded to the nearest integer, ignoring sub-grid point translations. $\Psi$ can be calculated for all time steps in which the storm is active, yielding the total smoothed swath area $\mathbb{S}$:

$$\mathbb{S} = \max\left(\left[\Psi^n\right]_{n\in\gamma}\right), \tag{A2}$$

where $\gamma$ contains all time steps where the storm is active. The described implementation bridges the gaps left by the sparse temporal sampling of the fast-moving, small hailstorms, while conserving small-scale details, even in the presence of complex
features, like crossing tracks, bowed squall lines, and multiple maxima within storm extents (Fig. A2b). With gap filling ($\kappa = 5$), the swath areas ($> 20\,\mathrm{mm}$) are a factor two larger than the swaths reported using the original 5 min time step (Fig. A2c).

## Appendix B: Single storm cross-section

In addition to the storm-centered composites shown in Fig. 5, we provide a selected vertical cross-section through a selected storm at its most intense time step (Fig B1).

## Appendix C: Ensemble spread of composites

As an addition to Fig. 5, the ensemble spread of selected variables provides context for the composite approach (Fig. C1).

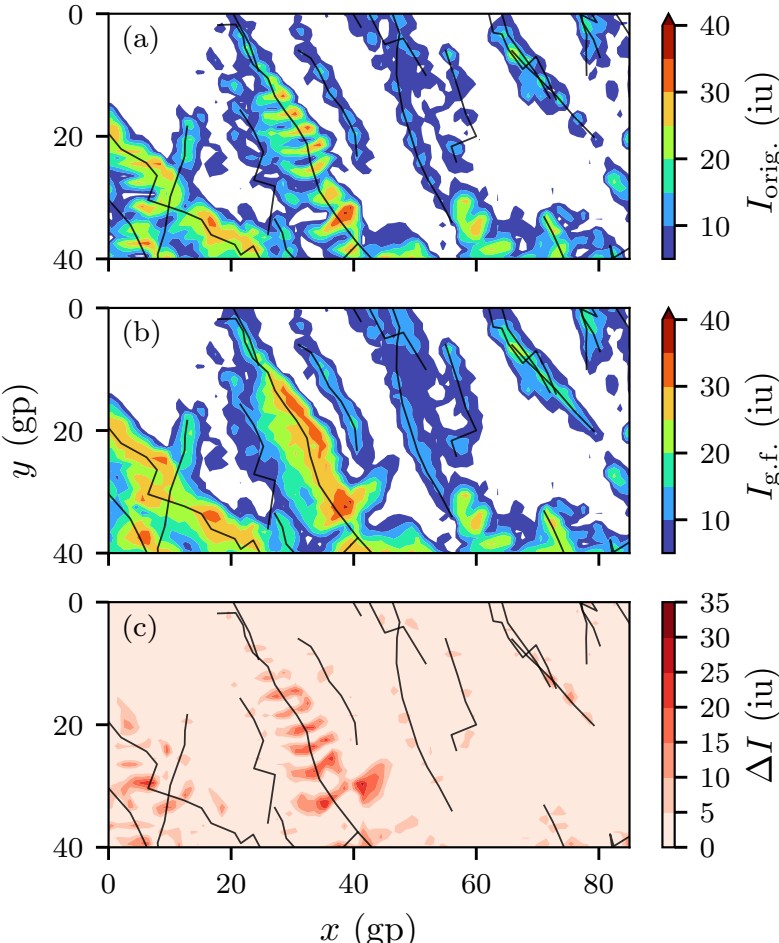

**Figure A2.** (**a**) Time-maximum original intensity field visualized as filled contours with storm tracks overlaid as black lines. (**b**) Identical to panel **a** but with gap filling enabled. (**c**) Intensity added by gap filling, or difference in intensity between panels **a** and **b**.

*Author contributions.* During the extensive development of this study, all coauthors contributed valuable comments and suggestions through in-depth discussions. Furthermore, specific contributions include **KB**: conceptualization of the study, development of the methodology and tracking algorithm, visualizations and manuscript writing, review, and editing. **MS**: methodology, review, and editing. **AW**: computation of the simulations, review and editing. **MA**: review and editing. **HW**: conceptualization, review, and editing.

*Competing interests.* At least one of the (co-)authors is a member of the editorial board of *Weather and Climate Dynamics*.

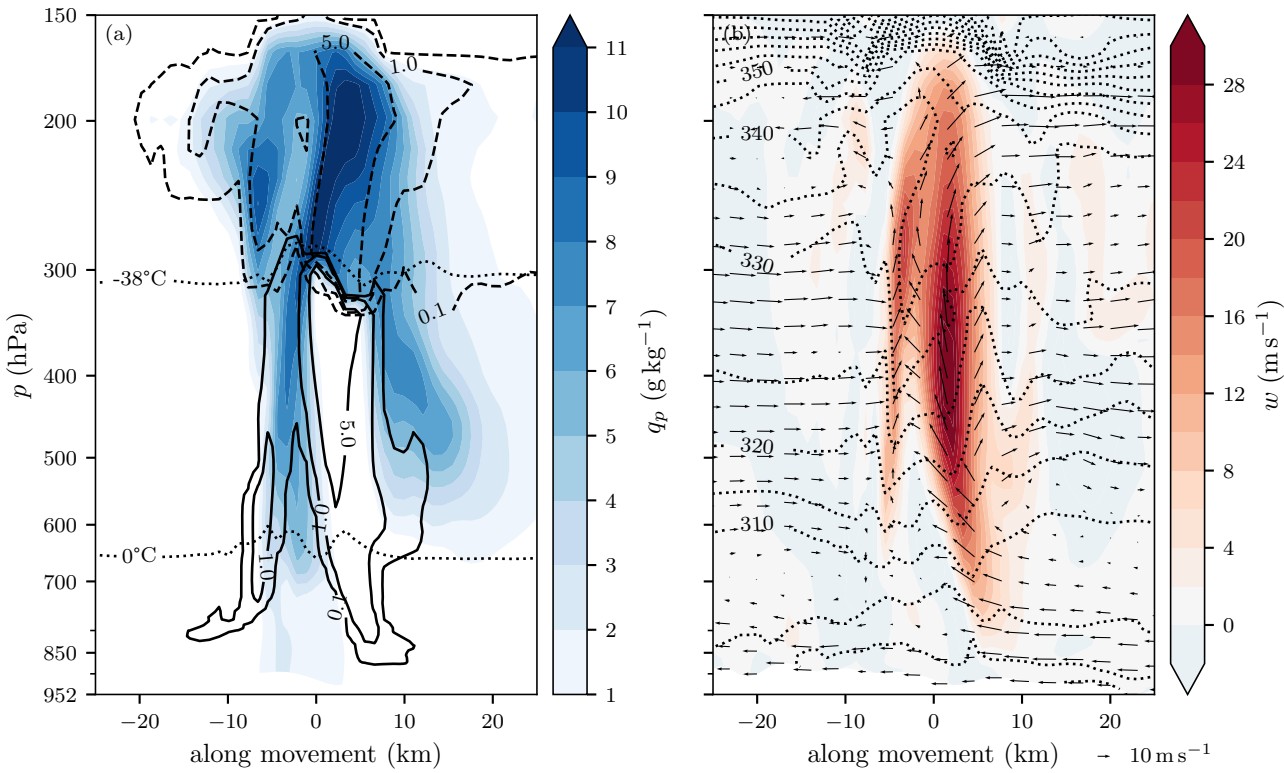

**Figure B1.** Same as Fig. 5, for the most intense time step of a selected storm in one member of the COSMO-1E simulation.

*Acknowledgements.* We extend our gratitude to the colleagues from ETH, MeteoSwiss, and the entire scClim team (https://scclim.ethz.ch/) for their valuable inputs and discussions. Further thanks go to Michael Blanc for reviewing the tracking algorithm. This study was funded by the Swiss National Science Foundation (SNSF) Sinergia grant `CRSII5_201792`. We acknowledge the use of OpenAI's GPT in assisting with language refinement in the preparation of this manuscript.

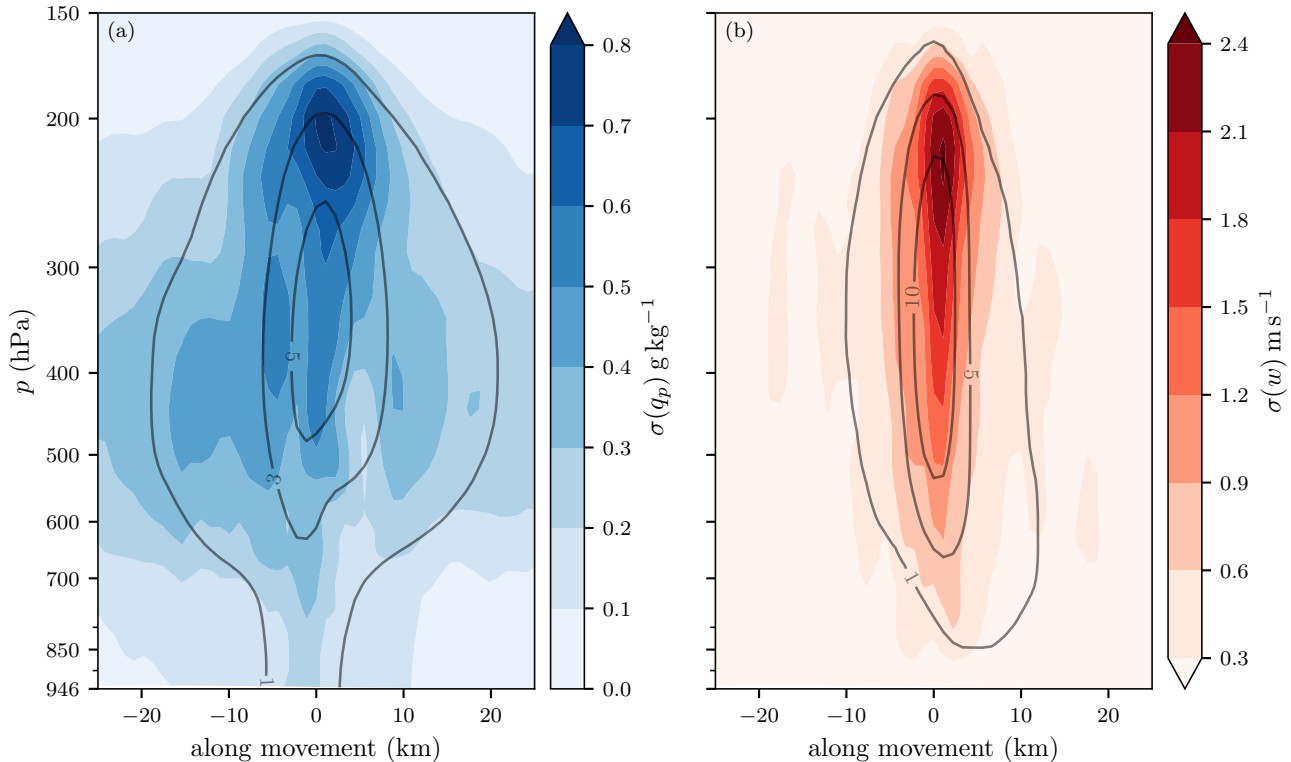

**Figure C1.** Adapted from Fig. 5, filled contours show the standard deviation across ensemble members, while the solid contours show the mean values for (**a**) precipitating hydrometeors, and (**b**) vertical wind. Storm-centered composite of $n = 100$ storms with lifespans $> 2.5\,\mathrm{h}$ at time steps with $w > 25\,\mathrm{m\,s^{-1}}$, identified in all COSMO-1E ensemble members.

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
