# Peer review of "An object-based and Lagrangian view on an intense hailstorm day in Switzerland as represented in COSMO-1E ensemble hindcast simulations"

_EGUsphere, 2024_

## Referee Comment (RC1)

Review: EGUSphere 2012-2148

Recommendation: Reject

This article uses a newly developed storm updraft tracking algorithm to track long-lived cells in an ensemble simulation of a day when a lot of hailstorms were experienced in Switzerland. The tracked cells are compared to radar characteristics, and mean cell and environmental characteristics are calculated. Finally, Lagrangian tracers are used within the cells to examine storm inflow over time.

I spent a lot of timing thinking about this paper when conducting my review. There is clearly a lot of quality work that was done for this project. Yet, I was concerned as each new conclusion was drawn: I kept feeling that more analysis was needed to support said conclusion. In the end, I'm not certain I'm fully comfortable with any of the drawn conclusions without inclusion of more analysis. But how to square that with how much work went into this paper?

After much thought, I believe the problem is the paper is trying to do too many things. There could be several papers' worth. I list each below and add what additional analysis would need to be included.

1. Methodology paper explaining the cell tracking algorithm and example applications
   o Add: Comparison of results with existing tracking algorithms. There are several existing tracking algorithms that have dynamical time tracking and splitting/merging (PyFlexTRKR as one example). Why was development of this algorithm necessary?

2. Examination of tracked cell characteristics in ensemble and comparison to radar observations, exploring the physical processes behind the differences.
   o Add: Track the cells in model data using reflectivity, not updraft speed, so the radar and model tracks can be compared 1-to-1.
   o Why does the ensemble underproduce long-lived cells, and very large/very small cells? What are the microphysical and/or dynamical characteristics of the cells it tends to get right, and those it misses?
   o Why does the ensemble produce a lot of cells in eastern France that weren't observed via radar? Why does it largely miss the cells in northern Italy (assuming that's not a radar gap issue)?

3. Use of model simulation to understand physical processes ongoing in the 28 June 2021 hailstorm in Switzerland. This storm is the only one with extensive surface hail observations reported in Kopp et al.; surface hail verification is highly necessary for reasons described below.
   o Add: Full verification of the modeled hail swath and radar reflectivity presentation. This is particularly important because HAILCAST is a model diagnostic (see major comment #1 below) and not explicitly produced by hail-related physical processes in the model. Kopp et al. is an excellent resource of information about the hail swath produced by this storm. How does the modelled HAILCAST hail swath compare? Is the model simulation representative of what happened? Does the convective mode of the storm and its structure appear similar in the model and radar observations? Only if so am I comfortable with using the subsequent model analysis to determine the underlying physical processes important to hail production. If the simulation is not representative of observations, the conclusions drawn about those physical processes could be incorrect.
   o Trimming the analysis to one cell instead of many will also solve the problem of many fine-scale storm (and storm-surrounding) structures being smoothed and averaged out in the environmental field analysis. This smoothing is particularly

concerning if it is being conducted across multiple convective modes and/or storm types.

- o Trimming subjects that fit under paper ideas 1 and 2 above (cell tracking algorithm, cell characteristics) will also allow for connection of this analysis to previous studies in the literature that examine the impact of storm-scale environmental flows on hailstorm development (e.g., Dennis and Kumjian 2017; Kumjian and Lombardo 2020; Lin and Kumjian 2022). The Lagrangian parcel technique the authors are using offer a novel way of exploring if the results of these idealistic studies can be seen or applied in a real-data simulation.

I would be happy to review a revised version of this paper (or multiple papers) that follows one or more of these suggestions. As the paper stands, however, I'm not comfortable with its acceptance.

Major comments: (These assume a goal of a paper like idea #3 listed above)

- Lines 65- 67: Given the fine scale features that are important to both convective updrafts (Bryan et al. 2003), I'm concerned that a convective-permitting resolution simulation alone won't be enough to address the objectives the authors have laid out re: storm structures being resolved in the simulations, particularly updrafts and downdrafts. How do the authors plan to address this issues? Analyzing results in the context of existing idealized convective-resolving simulations and noting the limitations of the coarser convective-permitting simulation is one possibility.

- Section 2.2: It isn't clear to me why an on-the-fly feature tracking algorithm like those described in Lines 131 -133 can't just be used here; it seems like it would have saved a lot of development trouble- why wasn't it?
  That being said, the whole paragraph from Lines 131-147 could be trimmed down to almost a single sentence saying that tracking requirements include the ability to handle coarser time resolution (hence dynamical tracking) and splitting/merging, which inspired the authors to create their own. The tracking algorithm isn't the point of the paper.

- The introduction is a mix of too much and yet not enough information. The purpose of ensembles (e.g., Lines 39-47) is already well-established and doesn't need to be explained. However, the authors do need to include discussion of existing research into how airflow around a storm impacts the storm and hail trajectories within it. There is some (very) small discussion on lines 72-74, but this should be expanded. Note the Prein and Heymsfield study used reanalysis data, not convective-resolving simulations.

- Sections 2. 2, 3.2: In a study like this one, where the authors are hoping to understand physical processes using a model simulation, it is important to determine how (or even if) the simulation is representative of the observations.

  - If thinking of paper idea #2: Verification is obviously difficult to do for something as transitory as deep convection, but a comparison of tracked cell lifespans and areas to observed values seemed promising. However, I am quite confused why the simulation data would be tracked via vertical velocity, a field with no observational equivalent. Why not track both the observed and simulated reflectivity, and then all the cell track information can be compared me-to-one?

  - Again, for paper idea #2 Lines 201-202: I don't know that I'd consider these runs to be a good representation given what is shown. Fig. 3 makes it look like the model runs way over-produced convective cells (although that's partially due to comparing all ensemble members to radar. Can the individual members be colored separately?)

However, if the goal is to successfully reproduce the hail swath in red, that looks like it was done by many members.

- I had a lot of trouble following if the results in Sections 4 and 5 were from just one cell or all tracked cells. The text seemed to switch back and forth frequently. I also had trouble discerning if the averaging was being conducted in time, along-track, cross-track, over multiple cells, and/or over multiple ensemble members. That's a lot of potential variability to keep straight! Honestly, given the importance of fine-scale features in hail production, I don't see what is gained by averaging the results across multiple tracked cells together (hence why I recommend focusing on just one cell). I'd like to see some standard deviation information included in Fig. 5 to provide needed context from the averages over time and/or ensemble members.

- Line 329-336: I would be careful with this analysis. HAILCAST is only a diagnostic tool, and its output shouldn't be used to explore scientific processes behind hailfall generation. Its 1d nature means it won't be able to take advantage of increased updraft width as important for hail growth, which has been established as important for a while (e.g., Nelson 1983, 1987). Thus any correlation between environmental parameters and HAILCAST hail size will be suspect. If the authors chose to carefully compare the HAILCAST size to observed hail reports, and found them to strongly correlate, then I would be more comfortable with relating HAILCAST hail size with environmental model variables.

Minor comments:

- Section 2.1.1 The version of HAILCAST you are using here is best termed "CAM-HAILCAST", or convective-allowing model-HAILCAST, to differentiate it from the HAILCAST of Brimelow and Jewell & Brimelow. Brimelow-HAILCAST used a steady-state cloud model connected to a hail growth model, as you describe in lines 113-115. CAM-HAILCAST, used herein, embeds a pseudo-Lagrangian 1D hail growth model into a convection allowing model, and should be attributed to Adams-Selin and Ziegler (2016) and Adams-Selin et al. (2019).
  Additionally, while Brimelow-HAILCAST is a 1d model, CAM-HAILCAST is "pseudo-Lagrangian" as it parameterizes hailstone horizontal motion across the updraft by adding a time- dependent updraft multiplier term (essentially a cosine curve).

- Line 119: Multiple initial embryo sizes and at multiple temperatures.

- Line 115: The CAM-HAILCAST outputs are the max, mean, and standard deviation of the different hail sizes produced by the different embryos, not of the entire storm. It would be worth including a small correction.

- Line 120: If only the hail sizes from the 10-mm embryo are used, where do the $d_{hail}$ distributions come from in Fig. 7?

- Line 165: Minor quibble, but since we don't have observations of the hail swaths, I wouldn't say "covered" by hail. Perhaps event with reports from the most towns, or something like that.

- Line 176: "pressure distribution... was flat." What does this mean?

- Lines 178-179: Interesting that the hodograph is almost a straight line. How unidirectional was the shear? That's unusual for hailstorms, at least in the U.S. How do you expect this could have affected the storm morphology?

- Line 180: I don't know that I'd consider that profile to have a high level of moisture above (colder than) 0°C.

- Lines/80-185, Fig. 1b: Why is this profile chosen? How representative is it of the wider area over which hailstorms initiated?

- Lines 196-197, Fig. 2b: Interesting result. What would you say this means physically?

- Line 214: Is this hail diameter produced from just one ensemble member, or all of them? How do the hail sizes produced compare across storm lifetime, ensemble member, and to observed reports?

- Line 213, Sections 4 and 5: Is this analysis from just one storm, a several of them? Lines 218- 222 make it sound like several storms, but 225 says just one.

- Line 235: It is interesting that less intense phase correlates with a temporary increase in terrain height. I'm curious what the wind profile is doing at that time. Can you calculate a storm inflow parameter, based on the low-level winds and storm motion, to see how that is changing? Also, how does the storm structure change during this time? Is it (still) linear? How wide are the updrafts? There are plenty of places to examine for a reason why the max updraft is weaker.

- Fig. 4: I would replace pressure on the y axis of (a) and (b) with temperature, as it is more relevant to hail growth. You could also add a horizontal line where T = 0C to help guide the reader's eye.

- Line 239: 0-6km wind shear should be shown. Perhaps divide Fig. 4 into two figures. That would also give you more room to expand (a) and (b) horizontally. A colorbar with more distinction in it would help too.

- Lines 246- 249: I am wary of conducting an analysis on storms that were not observed on radar in real life. Could you at least also require the modeled storm tracks to have a similar, nearby observed radar track, and remove from the analysis those that don't? I'm also wary given the differences in maximum cell areas shown in Fig. 2b between modeled and observed. The model could have gotten the convective mode of some of the cells wrong, as it doesn't have enough small cells or very large cells compared to the radar values.

- Line 247: Are only storms that reach a W of at least 25 m/s considered, or only the timesteps where W is > 25 m/s? If the latter, how many time steps are included from each cell? Are there enough to be representative?

- Line 251, Fig. 5a: I wouldn't consider a figure with only two contours for each field (and averaged over numerous cases) to "reveal intricate cloud structures."

- Fig. 5a: Is "total hydrometeor mixing ratio" only the precipitating hydrometeors? If so, I would note that.

- Fig. 5: Are these vertical cross-sections averaged in the across-track direction at all?

- Lines 257-258: While I agree with you about the cloud ice, precipitation (assuming that's the color fill in Fig. 5b) forms much earlier and at warmer temperatures. You could say it reaches a maximum around -38°C.

- Lines 257-264: You're right that little cloud ice is being produced before -38°C, but that's not all the frozen liquid fields. I recommend reviewing the snow and graupel mixing ratio fields, which I expect will extend much lower.

- Lines 265- 266: But remember, averaging in time will act to smooth out fine-scale features, making them appear more like larger-scale subsidence. Standard deviation

would be helpful here as a start. It could also simply be that multiple storm modes (and downdraft locations) are being averaged together and cancel each other out (as explained in Lines 266-270.) I would remove the large-scale process supposition from the text unless more evidence can be provided.

- Line 273: There's a cold pool due to evaporative cooling in a convective downdraft. That's not likely to be caused by large-scale downward motion.

- Fig 6: Without any scale information in the circles, it's hard to tell if features across subfigures are collocated. On the other hand, I understand the desire to avoid cluttering the figures. I'd add a dot representing storm track center (and thus updraft center, right?) at least. Possibly also interior x and/or y axes.

- Lines 274 - 278: It's hard to tell, but I don't think the area of surface convergence (a) and updraft core (c) overlap.

- Section 4.2: Convective morphology will be important when calculating area-mean composites. Are all the storms cellular, or some linear? If they have internal rotation, are they rotating the same way?

- Line 282-238: It would be very interesting to know how much this updraft core width varies from cell to cell.

- Line 288-289: Yes, good point, but would rephrase to "horizontal advection of hailstones to other grid columns."

- Line 305: I thought only timesteps where updraft velocity was > 25 m/s were included?

- Line 309: "From this...": this what?

- Fig. 7: Please include some grid lines within the plot so the reader can more easily estimate magnitudes. Height of the maximum vertical velocity (or rather, temperature) would also be interesting to include.

- Lines 320-323: I'd argue this result is one of the key takeaways from this figure discussion, but it isn't shown! It should be (perhaps added to (d) ?)

- Lines 323-324, Fig. 7f: It isn't clear to me what the physical meaning behind this subfigure is.

- Line 344: Why z = 5000 m?

- Section 5: This section is back to a single storm track, correct? Just one ensemble member, and if so, which?

- Line 349: What time is its highest intensity?

- Line 345: Is this criteria used to ensure the air parcel is lifted within the updraft? How effective is it, and now necessary is it? How do parcels end up in the updraft core if not through the updraft base- entrainment? It would be interesting to see if the amount of environmental entrainment into the updraft changes over the course of the mature period.

- Fig 9: Minor quibble, but I'd reverse the direction of the colorbar so 750 hPa is at the top.

- Lines 350- 354: Really interesting work. What can you say abut the vertical distribution of the inflow trajectories over time? Does that result agree with the mean values calculated in lines 319- 324? The idea that storm-relative inflow becomes broader and less coherent as a storm moves toward dissipation is an intriguing one that should be called out for future research. A comparison to previous studies of storm-relative inflow of similar convective modes would be good here too.

- Fig. 10: Now back to all cells?
  Also, are these values averaged over all times, with t= 0 when the trajectory enters the updraft? Or are they relative to just the most intense time of each cell? I'm concerned if the increasingly broadening inflow shown in Fig. 9, for example, is being averaged all together in Fig. 10.

- Line 358: I see no plot of latent heat release (it could be calculated from the model data if you want it, perhaps?)

- Lines 383-384: Good result. The rain falling into the updraft could be a result of the "coarser" (not convective-resolving) nature of a 1.1-km grid: the updraft and downdraft are not sufficiently resolved so sometimes portions of them occur in the same grid cell.

- Line 417: not "convective-resolving" (typically considered to be on the order of ~100 m, Bryan et al. 2003) but instead convective- permitting.

- Line 432: But the research results gleaned herein will still be subject to any internal biases inherent in the hail diagnostic.

---

## Author Comment (AC1)

EGUSPHERE-2024-2148

**An object-based and Lagrangian view on an intense hailstorm day in Switzerland as represented in COSMO-1E ensemble hindcast simulations**

Killian P. Brennan, Michael Sprenger, André Walser, Marco Arpagaus, and Heini Wernli

**Final author comments**

We thank the two reviewers and the editor for their insightful and helpful comments. We are aware that we received strongly contrasting evaluations of our paper, with reviewer 1 suggesting to reject the paper (but still mentioning that "there is clearly a lot of quality work that was done for this project" and eventually rating the paper as "fair" in terms of scientific relevance, "good" in terms of "scientific quality, and "excellent" in terms of presentation quality) and reviewer 2 being overall very positive (rating the paper as "excellent" in all categories). The editor's assessment was again more critical mentioning that "the paper seems to state a lot of facts about the simulation without a clear story or any in-depth analysis". This clearly implies that we have to improve the storyline of our paper and better explain our main findings. Before we address the specific points mentioned by the reviewers and the editor, we discuss a few important and general aspects of our study that need improvement.

Critique about missing story (editor and reviewer 1): We agree with the need to improve the outline of our paper and better explain our main goals (at the end of the introduction) and main findings (in the conclusions). We realized that in particular reviewer 1 obviously expected a different paper after reading the introduction than what we actually presented. We should also make clear what our paper is *not* meant to be: (i) we use a newly developed hailstorm tracking (tailored specifically for our simulations), which we regard as an essential tool for our life cycle composite analyses, but the paper is not meant to show that this tracking is better than other tracking approaches; (ii) we certainly cannot validate an ensemble simulation model with a single case study; this is fundamentally impossible, and therefore we cannot follow the guidance of reviewer 1 to first validate our model before investigating hailstorm tracks[1]; and (iii) we are fully aware that this is "only" a case study of one particular (really intense) hailstorm day in Switzerland, and as we mentioned in the original paper, it remains difficult to draw general conclusions outside the specific case. However, we still regard our paper as original and important. **Our approach is to use the operational ensemble simulations by MeteoSwiss of this hailstorm day as a "free set of sensitivity experiments".** Given the design of the ensemble, each ensemble member is physically consistent and equally likely, and by regarding hailstorm tracks in all ensemble members, we can increase our statistics and obtain more robust findings about, for instance, the track environment. We will make this point clearer in the revised version of
* * *
[1] The group has experience in ensemble validation studies, considering flow-dependent extended spread-error relationships (Rodwell and Wernli 2023), but this requires a much larger set of ensemble simulations, which in this case (simulations with 5-min 3-dimensional output from COSMO-1E) is currently not realistically available.

our paper. About our main findings, we would like to mention the following points. In our study, we show that

1) COSMO-1E with HAILCAST realistically simulates hailstorm tracks compared with radar data;

2) combined Eulerian and Lagrangian analyses reveal detailed hailstorm structures and dynamics (i.e. revealing the life cycle of an archetypical hailstorm from the studied environment); and

3) air parcel trajectories show that air entering the storm updraft originated from different environments (including air that crossed the Alps) and that it experienced rain before entering the updraft, which potentially moistened the inflow.

While point 1) will remain qualitative (as we cannot do a validation of an ensemble with a single case) and is in line with results from COSMO-2 simulations by Cui et al. (2023), we think that our findings 2 and 3 are interesting to the hail community and contain sufficient novelty that warrants publication. Of course, with the important caveat that our findings are based on simulations with a specific model, meaning that if in the future similar studies can be done with "better models" (e.g., with explicit hail microphysics), some findings might require modifications. To the best of our knowledge, our study is the first to study air parcel trajectories in a time-evolving severe convective system, supporting the novelty of point 3.

We address each comment point by point below. The reviewers' and editor's comments are given in blue and our responses are in black. In the revised manuscript, we plan to improve in particular the following aspects:

a) clarify the storyline and research focus (refocus the manuscript by streamlining the objectives and providing clearer scientific questions that guide the analysis);
b) provide more in-depth analysis of key findings and discuss their novelty and relationship to previous literature (and we plan, as suggested, to include a discussion of storm-relative helicity);
c) address trajectory sensitivity and robustness: Concerns were expressed about the sensitivity of the trajectories and the representativeness of the trajectories with origins south of the Alps. With our approach of launching ensembles of trajectories, we can quantify and discuss their sensitivity, including those that terminate south of the Alps.

**Reviewer 1**

Recommendation: Reject

This article uses a newly developed storm updraft tracking algorithm to track long-lived cells in an ensemble simulation of a day when a lot of hailstorms were experienced in Switzerland. The tracked cells are compared to radar characteristics, and mean cell and environmental characteristics are calculated. Finally, Lagrangian tracers are used within the cells to examine storm inflow over time.

I spent a lot of timing thinking about this paper when conducting my review. There is clearly a lot of quality work that was done for this project. Yet, I was concerned as each new conclusion was drawn: I kept feeling that more analysis was needed to support said conclusion. In the end, I'm not certain I'm fully comfortable with any of the drawn conclusions without inclusion of more analysis. But how to square that with how much work went into this paper?

After much thought, I believe the problem is the paper is trying to do too many things. There could be several papers' worth. I list each below and add what additional analysis would need to be included.

We are grateful for your thorough and thoughtful review of our manuscript. You provided valuable insights, particularly regarding the need for a more focused and in-depth analysis to support the conclusions presented. We acknowledge your concerns about the breadth of topics covered in the manuscript and agree that a more streamlined approach would allow for a clearer, more robust narrative that better ties together the different aspects of this study. We address your paper ideas in the following paragraphs, and by implementing your major and minor comments we think that the manuscript will be much improved.

1. Methodology paper explaining the cell tracking algorithm and example applications.

- Add: Comparison of results with existing tracking algorithms. There are several existing tracking algorithms that have dynamical time tracking and splitting/merging (PyFlexTRKR as one example). Why was development of this algorithm necessary?

In our view, a pure methodology paper explaining the cell tracking algorithm and example applications would be a less valuable contribution to the literature than the current manuscript, which provides a comprehensive analysis of the storm environment and physical processes. We believe that the current description of the tracking methodology as provided in the appendix is sufficient for the purpose of the manuscript and that the focus should remain on the scientific analysis. We will however extend the description of the algorithm in the appendix to address your concerns. Further, we will compare the tracks that were identified by our algorithm to a benchmark tracking algorithm (i.e., PyFlexTRKR) to assess the quality of our tracking algorithm.

2. Examination of tracked cell characteristics in ensemble and comparison to radar observations, exploring the physical processes behind the differences.

- Add: Track the cells in model data using reflectivity, not updraft speed, so the radar and model tracks can be compared 1-to-1.

- Why does the ensemble underproduce long-lived cells, and very large/very small cells? What are the microphysical and/or dynamical characteristics of the cells it tends to get right, and those it misses?

- Why does the ensemble produce a lot of cells in eastern France that weren't observed via radar? Why does it largely miss the cells in northern Italy (assuming that's not a radar gap issue)?

The first part of this suggestion was the aim of this paper (i.e., to examine the tracked cell characteristics in the ensemble). The comparison with observations (i.e., validation) is not straightforward for single case studies. This is discussed in detail in the following paragraph, see also main finding 1) above. In short, addressing your questions in the bullet points would require a more statistical analysis with forecasts for many hail events, which is not the aim of this study.

3. Use of model simulation to understand physical processes ongoing in the 28 June 2021 hailstorm in Switzerland. This storm is the only one with extensive surface hail observations reported in Kopp et al.; surface hail verification is highly necessary for reasons described below.

- Add: Full verification of the modeled hail swath and radar reflectivity presentation. This is particularly important because HAILCAST is a model diagnostic (see major comment #1 below) and not explicitly produced by hail-related physical processes in the model. Kopp et al. is an excellent resource of information about the hail swath produced by this storm. How does the modelled HAILCAST hail swath compare? Is the model simulation representative of what happened? Does the convective mode of the storm and its structure appear similar in the model and radar observations? Only if so am I comfortable with using the subsequent model analysis to determine the underlying physical processes important to hail production. If the simulation is not representative of observations, the conclusions drawn about those physical processes could be incorrect.

- Trimming the analysis to one cell instead of many will also solve the problem of many fine-scale storm (and storm-surrounding) structures being smoothed and averaged out in the environmental field analysis. This smoothing is particularly

- concerning if it is being conducted across multiple convective modes and/or storm types.
  Trimming subjects that fit under paper ideas 1 and 2 above (cell tracking algorithm, cell characteristics) will also allow for connection of this analysis to previous studies in the literature that examine the impact of storm-scale environmental flows on hailstorm development (e.g., Dennis and Kumjian 2017; Kumjian and Lombardo 2020; Lin and Kumjian 2022). The Lagrangian parcel technique the authors are using offer a novel way of exploring if the results of these idealistic studies can be seen or applied in a real-data simulation.

In our opinion, it is not meaningful to verify a single event in an ensemble simulation, as the analysis might find that the ground truth either falls within the ensemble spread or not (see also main finding 1) above). Verification of an ensemble prediction always requires large statistics for many events. In addition, results about a detailed comparison of observations vs. predictions would not be very informative when

considering only one event. A comprehensive validation of several seasons' worth of simulations is planned but falls outside the scope of this study. We argue, that addressing the "verification" through qualitative comparative analysis of the tracks discussed in Section 3.2 and arguing that the model can produce similar tracks to the observed ones (Fig. 3) is sufficient to apply the model output to process understanding.

A further major comment from you concerns our use of composite analysis, rather than investigating single storms. To address this, we will extend our single storm section (4.1) while also addressing comments from the editor. We are however of the understanding that the composite analysis allows us to determine properties of the storms that are more generally valid than what would be possible when studying a single hail cell where some signals might be hidden by the stochasticity of the convective environment. We will clarify this in the revised manuscript. Composite analysis is a widely established methodology to investigate synoptic systems (i.e. Catto et al., 2010, Binder et al., 2016) and has been recently applied to convective systems and storms (e.g. Prein et al., 2017, Oertel et al., 2020, and Lin and Kumjan, 2022). We will include this literature review on composite analysis in the revised manuscript.

Thank you for making us aware of the Kumjian et al. papers. Except for Lin and Kumjian (2022), they don't focus on air parcel trajectories, but rather on hail (embryo) trajectories. However, in Lin and Kumjian (2022), the approach towards computing trajectories is based on a steady-state composite of the convective system, which is distinctly different from the air parcel trajectories in the time-evolving flow used in our study. We will aim to better relate our findings with the results from those papers in the revised manuscript.

I would be happy to review a revised version of this paper (or multiple papers) that follows one or more of these suggestions. As the paper stands, however, I'm not comfortable with its acceptance.

Major comments: (These assume a goal of a paper like idea #3 listed above)

Lines 65- 67: Given the fine scale features that are important to both convective updrafts (Bryan et al. 2003), I'm concerned that a convective-permitting resolution simulation alone won't be enough to address the objectives the authors have laid out re: storm structures being resolved in the simulations, particularly updrafts and downdrafts. How do the authors plan to address this issues? Analyzing results in the context of existing idealized convective-resolving simulations and noting the limitations of the coarser convective-permitting simulation is one possibility.

Thank you for sharing this concern. We will include your considerations in the limitations section (lines 426 and following) and extend the discussion on the resolution limitations with Bryan et al. (2003), and Jeevanjee (2017).

Section 2.2: It isn't clear to me why an on-the-fly feature tracking algorithm like those described in Lines 131 -133 can't just be used here; it seems like it would have saved a lot of development trouble- why wasn't it?
That being said, the whole paragraph from Lines 131-147 could be trimmed down to

almost a single sentence saying that tracking requirements include the ability to handle coarser time resolution (hence dynamical tracking) and splitting/merging, which inspired the authors to create their own. The tracking algorithm isn't the point of the paper.

Thank you for this input. We provide reasoning behind developing our own algorithm on lines 148/149. As you suggest, we will shorten the paragraph on lines 131-133. And briefly about the on-the-fly tracking: such approaches are very challenging to implement, in particular in an operational environment as used here (recall that the COSMO-1E simulation we used corresponds to a rerun of the operational forecast by MeteoSwiss). We regard this as a vision for the future, but it was way beyond what we could implement in our project.

The introduction is a mix of too much and yet not enough information. The purpose of ensembles (e.g., Lines 39-47) is already well-established and doesn't need to be explained. However, the authors do need to include discussion of existing research into how airflow around a storm impacts the storm and hail trajectories within it. There is some (very) small discussion on lines 72-74, but this should be expanded. Note the Prein and Heymsfield study used reanalysis data, not convective-resolving simulations.

Thank you for mentioning this imbalance. We will trim the explanation of ensembles, and expand the discussion of the existing research into airflow around storms and hail trajectories within. We meant to cite Prein et al. (2017) in line 74 and will correct this in the revised manuscript.

Sections 2. 2, 3.2: In a study like this one, where the authors are hoping to understand physical processes using a model simulation, it is important to determine how (or even if) the simulation is representative of the observations.

Please refer back to the general remarks made about validation in the beginning of this document.

If thinking of paper idea #2: Verification is obviously difficult to do for something as transitory as deep convection, but a comparison of tracked cell lifespans and areas to observed values seemed promising. However, I am quite confused why the simulation data would be tracked via vertical velocity, a field with no observational equivalent. Why not track both the observed and simulated reflectivity, and then all the cell track information can be compared me-to-one?

For tracking the dynamic center of the storm, it is more physically meaningful to track the vertical wind field. The (simulated) reflectivity field and the hydrometeors that contribute to it are subject to advection, leading to fuzzier fields with less well-defined structures. The advantage of using the vertical wind as a track basis is especially beneficial to the composite analysis (where the storms will be better aligned) and when starting trajectories in the updraft. As we were not limited to having to choose an observable field to perform our tracking on, we decided on the vertical wind as our tracking basis. Of course, this is sub-optimal for the comparison with measurements,

however, this is not the main point of our study (see discussion at the beginning of this document).

Again, for paper idea #2 Lines 201-202: I don't know that I'd consider these runs to be a good representation given what is shown. Fig. 3 makes it look like the model runs way over-produced convective cells (although that's partially due to comparing all ensemble members to radar. Can the individual members be colored separately?)
However, if the goal is to successfully reproduce the hail swath in red, that looks like it was done by many members.

Thank you for sharing this concern. To address this, we will color each ensemble member separately in Fig. 3.

I had a lot of trouble following if the results in Sections 4 and 5 were from just one cell or all tracked cells. The text seemed to switch back and forth frequently. I also had trouble discerning if the averaging was being conducted in time, along-track, cross-track, over multiple cells, and/or over multiple ensemble members. That's a lot of potential variability to keep straight! Honestly, given the importance of fine-scale features in hail production, I don't see what is gained by averaging the results across multiple tracked cells together (hence why I recommend focusing on just one cell). I'd like to see some standard deviation information included in Fig. 5 to provide needed context from the averages over time and/or ensemble members.

Thank you for pointing this out. We aimed to introduce every new analysis concept with a single storm for illustrative purposes and then extend the analysis to all 100 storms for more generally valid results. However, we see that this led to confusion about what data the analysis was based on.

We will make sure that in each figure and each result, it can be unambiguously discerned whether a single or all 100 storms were used for the analysis.

The reason we did not focus more on single-cell analysis is that we want to present results that are more generally valid. As addressed earlier, the use of composite analysis for convective storms is well established (e.g. Prein et al., 2017, Oertel et al., 2020, and Lin and Kumjian, 2022) as we will mention in the revised manuscript.

Through a bootstrapping approach, we will determine the robustness of Fig. 5.

Line 329-336: I would be careful with this analysis. HAILCAST is only a diagnostic tool, and its output shouldn't be used to explore scientific processes behind hailfall generation. Its 1d nature means it won't be able to take advantage of increased updraft width as important for hail growth, which has been established as important for a while (e.g., Nelson 1983, 1987). Thus any correlation between environmental parameters and HAILCAST hail size will be suspect. If the authors chose to carefully compare the HAILCAST size to observed hail reports, and found them to strongly correlate, then I would be more comfortable with relating HAILCAST hail size with environmental model variables.

Thank you for addressing this. We will include a precise discussion of the limitations that arise from the HAILCAST diagnostic in this section of the revised manuscript.

Minor comments:

Section 2.1.1 The version of HAILCAST you are using here is best termed "CAM-HAILCAST", or convective-allowing model-HAILCAST, to differentiate it from the HAILCAST of Brimelow and Jewell & Brimelow. Brimelow-HAILCAST used a steady- state cloud model connected to a hail growth model, as you describe in lines 113-115. CAM-HAILCAST, used herein, embeds a pseudo-Lagrangian 1D hail growth model into a convection allowing model, and should be attributed to Adams-Selin and Ziegler (2016) and Adams-Selin et al. (2019).

Thank you for pointing out the inaccuracies in our HAILCAST description, we will change the paragraph accordingly and include the suggested references.

Additionally, while Brimelow-HAILCAST is a 1d model, CAM-HAILCAST is "pseudo-Lagrangian" as it parameterizes hailstone horizontal motion across the updraft by adding a time- dependent updraft multiplier term (essentially a cosine curve).

Thank you, we'll include this detail in the revised manuscript.

Line 119: Multiple initial embryo sizes and at multiple temperatures.

Thank you for this comment, we'll modify the sentence accordingly.

Line 115: The CAM-HAILCAST outputs are the max, mean, and standard deviation of the different hail sizes produced by the different embryos, not of the entire storm. It would be worth including a small correction.

As mentioned on line 120, we only use the largest, 10 mm embryo and thus only use the DHAILMAX variable from HAILCAST. The other variables were not mentioned, as we did not include them in our analysis.

Line 120: If only the hail sizes from the 10-mm embryo are used, where do the dhail distributions come from in Fig. 7?

This is the distribution of the hail diameters of the 100 storms investigated.

Line 165: Minor quibble, but since we don't have observations of the hail swaths, I wouldn't say "covered" by hail. Perhaps event with reports from the most towns, or something like that.

We are referencing Kopp et al. (2022) here, who estimated the hail area from the radar variable MESHs. We'll change the sentence to: "...largest area within Switzerland affected by severe hail...".

Line 176: "pressure distribution... was flat." What does this mean?

We will discuss the meaning of a flat pressure distribution in the revised manuscript (it refers to weak horizontal gradients).

Lines 178-179: Interesting that the hodograph is almost a straight line. How unidirectional was the shear? That's unusual for hailstorms, at least in the U.S. How do you expect this could have affected the storm morphology?

Thank you for noticing this. Since the hodograph is almost a straight line, the shear is unidirectional. Unlike in the U.S., hail days in Europe seem to feature more straight-line hodographs (high SRH values are anyway rare). Unfortunately, this is only anecdotal, and a detailed investigation out of the scope of our study.

Line 180: I don't know that I'd consider that profile to have a high level of moisture above (colder than) 0°C.

Thank you for this observation, we'll change the sentence to: "The vertical profile also reveals a moderate level of moisture…".

Lines/80-185, Fig. 1b: Why is this profile chosen? How representative is it of the wider area over which hailstorms initiated?

Payerne is the site chosen by the Swiss weather service for their operational balloon soundings. It was carefully chosen (by MeteoSwiss) to provide measurements that are representative of the wider North-Alpine area of Switzerland. In effect, it is also the only station in Switzerland with regular soundings.

We chose to show the simulated profile at the same location, so that we can directly compare it to the measured profile.

Lines 196-197, Fig. 2b: Interesting result. What would you say this means physically?

We will discuss the potential causes for this discrepancy in the revised manuscript.

Line 214: Is this hail diameter produced from just one ensemble member, or all of them? How do the hail sizes produced compare across storm lifetime, ensemble member, and to observed reports?

This is from just one member, as indicated in the preceding sentence. We will include an additional panel in Fig. 8 that shows the relation between storm lifetime and hail diameter.

Line 213, Sections 4 and 5: Is this analysis from just one storm, a several of them? Lines 218- 222 make it sound like several storms, but 225 says just one.

Thank you for this comment. To make this sentence less ambiguous, we will change the sentence to: "Preceding the analysis of all selected storms, we introduce the different analysis concepts in Sect. 4 and 5 with detailed analyses of an individual hail cell. To this end, one exemplary storm was selected from ensemble member 5 …"

Line 235: It is interesting that less intense phase correlates with a temporary increase in terrain height. I'm curious what the wind profile is doing at that time. Can you calculate a storm inflow parameter, based on the low-level winds and storm motion, to see how that is changing? Also, how does the storm structure change during this time? Is it (still) linear? How wide are the updrafts? There are plenty of places to examine for a reason why the max updraft is weaker.

We will include a plot of the updraft in Fig. 4 and discuss the reasoning behind the weaker max updraft during this less intense phase.

Fig. 4: I would replace pressure on the y axis of (a) and (b) with temperature, as it is more relevant to hail growth. You could also add a horizontal line where T = 0C to help guide the reader's eye.

Thank you for this suggestion. We will add temperature isolines to the first two panels of Fig. 4.

Line 239: 0-6km wind shear should be shown. Perhaps divide Fig. 4 into two figures. That would also give you more room to expand (a) and (b) horizontally. A colorbar with more distinction in it would help too.

We will include an additional panel with vertical 0-6 km shear in Fig. 4.

Lines 246- 249: I am wary of conducting an analysis on storms that were not observed on radar in real life. Could you at least also require the modeled storm tracks to have a similar, nearby observed radar track, and remove from the analysis those that don't? I'm also wary given the differences in maximum cell areas shown in Fig. 2b between modeled and observed. The model could have gotten the convective mode of some of the cells wrong, as it doesn't have enough small cells or very large cells compared to the radar values.

Thank you for sharing your concern. We discussed this in the general remarks at the beginning of this document, concerning the "free set of sensitivity experiments".

Line 247: Are only storms that reach a W of at least 25 m/s considered, or only the timesteps where W is > 25 m/s? If the latter, how many time steps are included from each cell? Are there enough to be representative?

As the wording "… at time steps with …" suggests it's the former. Across the 100 storms, 2261 time steps are included in total in the composite, this will be mentioned in the revised manuscript.

Line 251, Fig. 5a: I wouldn't consider a figure with only two contours for each field (and averaged over numerous cases) to "reveal intricate cloud structures."

Thanks, we'll change the sentence to "…reveal the cloud structures".

Fig. 5a: Is "total hydrometeor mixing ratio" only the precipitating hydrometeors? If so, I would note that.

Thank you for this comment, we'll note that.

Fig. 5: Are these vertical cross-sections averaged in the across-track direction at all?

No, the values shown in the cross sections are from a single plane. We will change the caption to "Vertical cross-section (single plane through storm center) along the propagation direction..."

Lines 257-258: While I agree with you about the cloud ice, precipitation (assuming that's the color fill in Fig. 5b) forms much earlier and at warmer temperatures. You could say it reaches a maximum around -38°C.

We assume you are talking about Fig. 5a, as stated in the caption, the color fill in Fig. 5b is the vertical wind field. As also stated in the caption of Fig.5 "Filled contours denote the total hydrometeor mixing ratio...".

Well spotted, we will mention that the maximum is reached at -38°C.

Lines 257-264: You're right that little cloud ice is being produced before -38°C, but that's not all the frozen liquid fields. I recommend reviewing the snow and graupel mixing ratio fields, which I expect will extend much lower.

We will review those fields as suggested and discuss those findings in the revised manuscript.

Lines 265- 266: But remember, averaging in time will act to smooth out fine-scale features, making them appear more like larger-scale subsidence. Standard deviation would be helpful here as a start. It could also simply be that multiple storm modes (and downdraft locations) are being averaged together and cancel each other out (as explained in Lines 266-270.) I would remove the large-scale process supposition from the text unless more evidence can be provided.

We will remove the sentence "This points towards the possibility of large-scale subsidence as a process for balancing the storm updrafts, rather than localized precipitation-associated downdrafts." and the "However" in the following sentence.

We will however move the speculation toward the large-scale subsidence being the dominant means of balancing the storm updrafts further down where we reference Fig. D1 (line 270).

Line 273: There's a cold pool due to evaporative cooling in a convective downdraft. That's not likely to be caused by large-scale downward motion.

We will change the sentence to: "Both extrema are caused by the downdraft air entrained by intense precipitation and evaporative cooling forming a cold pool."

Fig 6: Without any scale information in the circles, it's hard to tell if features across subfigures are collocated. On the other hand, I understand the desire to avoid cluttering

the figures. I'd add a dot representing storm track center (and thus updraft center, right?) at least. Possibly also interior x and/or y axes.

Thank you for pointing this out. To address this we will add internal dotted or dashed x and y axes which will also indicate the center at their intersection.

Lines 274 - 278: It's hard to tell, but I don't think the area of surface convergence (a) and updraft core (c) overlap.

Many thanks for this comment. As seen in Fig. 5b, the updraft core is slanted, and we believe the sentence on lines 276/277 "The location of near-surface convergence of the horizontal wind denotes the beginning of the updraft column." reflects this adequately and does not suggest that the surface convergence and the updraft core position at 400 hPa should overlap. We can specify the paragraph by adding "Note that as the updraft core is slanted (Fig. 5b), near-surface convergence (Fig. 6a) does not overlap with the horizontal position of the updraft core aloft (Fig. 6c)."

Section 4.2: Convective morphology will be important when calculating area-mean composites. Are all the storms cellular, or some linear? If they have internal rotation, are they rotating the same way?

We are grateful that you share this concern with us. As the storms all arise from a similar environment, they are all of similar structure, we will discuss this here and include the fraction of cyclonic and anticyclonically rotating storms.

Line 282-238: It would be very interesting to know how much this updraft core width varies from cell to cell.

Thank you for finding this gap. We will determine the distribution of core widths and include the standard deviation here.

Line 288-289: Yes, good point, but would rephrase to "horizontal advection of hailstones to other grid columns."

Well spotted, this will be changed as suggested.

Line 305: I thought only timesteps where updraft velocity was > 25 m/s were included?

Thank you for asking this question. On line 190 we specify that we only investigate storms that exceed 25 ms$^{-1}$ throughout their life, however, as specified on line 305, the track of the storms starts when the more than 5 grid-points exceed 5 ms$^{-1}$.

Line 309: "From this...": this what?

Thank you for pointing out this inaccurate reference, will be changed to: "From the development-stage relative view, ...".

Fig. 7: Please include some grid lines within the plot so the reader can more easily estimate magnitudes. Height of the maximum vertical velocity (or rather, temperature) would also be interesting to include.

We will include grid lines in Fig. 7.

Lines 320-323: I'd argue this result is one of the key takeaways from this figure discussion, but it isn't shown! It should be (perhaps added to (d) ?)

Thank you for this comment, we will add a panel with the height above ground level to Fig. 7.

Lines 323-324, Fig. 7f: It isn't clear to me what the physical meaning behind this subfigure is.

A larger updraft area would tend to harbor more of the rapidly ascending trajectories. However, a direct physical meaning behind the magnitude of this quantity cannot be assigned, as it is a function of the starting grid density of the trajectories.

Line 344: Why z = 5000 m?

We did initial testing where we used multiple starting levels for the trajectories, however, since we are only looking at the rapidly ascending trajectories, they quickly pass through the levels we tested (200, 500, 1000, 2000, 4000, 8000). The problem however with using multiple levels for the start points is that the parcels will not be synchronized to a common time / height, thus making compositing more difficult.

Section 5: This section is back to a single storm track, correct? Just one ensemble member, and if so, which?

Correct, as stated in the response to one of your major comments, we aimed to introduce every new analysis concept with a single storm for illustrative purpose and then extend the analysis to all 100 storms for more generally valid results. Ensemble member 5 will be specified.

Line 349: What time is its highest intensity?

This detail will be added to the revised manuscript (14:35 UTC).

Line 345: Is this criteria used to ensure the air parcel is lifted within the updraft? How effective is it, and now necessary is it? How do parcels end up in the updraft core if not through the updraft base- entrainment? It would be interesting to see if the amount of environmental entrainment into the updraft changes over the course of the mature period.

Exactly, we are only interested in analyzing the rapidly ascending parcels in this study. We will include the fraction of trajectories that satisfy the filter criterion in the revised manuscript, thank you for this suggestion.

Fig 9: Minor quibble, but I'd reverse the direction of the colorbar so 750 hPa is at the top.

Thank you for pointing this out, this will be changed in the revised manuscript.

 Really interesting work. What can you say abut the vertical distribution of the inflow trajectories over time? Does that result agree with the mean values calculated in lines 319- 324? The idea that storm-relative inflow becomes broader and less coherent as a storm moves toward dissipation is an intriguing one that should be called out for future research. A comparison to previous studies of storm-relative inflow of similar convective modes would be good here too.

Thank you! This can be answered by the pressure in Fig. 10a. The trajectories remain close to the ground and only ascend in the last 30 min as discussed in the paragraph starting on line 355 and in agreement with statements in lines 319-324. We will include the suggested future research call-outs.

Fig. 10: Now back to all cells?
Also, are these values averaged over all times, with t= 0 when the trajectory enters the updraft? Or are they relative to just the most intense time of each cell? I'm concerned if the increasingly broadening inflow shown in Fig. 9, for example, is being averaged all together in Fig. 10.

As commented above, we aimed to introduce every new analysis concept with a single storm for illustrative purpose and then extend the analysis to all 100 storms for more generally valid results.

A note will be included in the caption to clarify that these are in fact values from all cells averaged over all cell times with $t$=0 when the trajectory enters the updraft. Your concern about the averaging over the broad inflow will be discussed.

Line 358: I see no plot of latent heat release (it could be calculated from the model data if you want it, perhaps?)

We will remove any mention of latent heat release here.

Lines 383-384: Good result. The rain falling into the updraft could be a result of the "coarser" (not convective-resolving) nature of a 1.1-km grid: the updraft and downdraft are not sufficiently resolved so sometimes portions of them occur in the same grid cell.

Thank you for this comment, we will discuss this limitation. However, in Fig. 10, it is evident, that the $Q_r$ is the first species to increase in concentration, even at -0.5 h away from reaching the storm, which would account for several grid-points in front of the storm.

Line 417: not "convective-resolving" (typically considered to be on the order of ~100 m, Bryan et al. 2003) but instead convective-permitting.

We will change the nomenclature according to your suggestion (lines 72, 100, and 417).

Line 432: But the research results gleaned herein will still be subject to any internal biases inherent in the hail diagnostic.

Thank you, this limitation will be added to the revised version of the manuscript.

**Reviewer 2**

Summary: This survey investigates the dynamical processes on an intense hailstorm day in Switzerland from Eulerian and Lagrangian points of view using high-resolution convection-permitting COSMO-1E ensemble simulations (Dx = 1.1km). Moreover, the authors use an object-based method to track the hailstorm cells. The authors compare the results obtained by the tracking in COSMO-1E simulations and the radar observations.

The manuscript is very well written, and it is a remarkable contribution to the knowledge of hailstorms in Europe and their interactions with complex orography. Although this manuscript could be divided into two papers due to the complexity of the research, the authors have managed correctly to display the information in the figures and text. In this way, this manuscript is easy to follow for the readers.

Therefore, this survey should be published in WCD with only a few technical corrections for better understanding by the readers.

We are grateful for your positive evaluation of our manuscript and your recognition of its contribution to the understanding of hailstorms in Europe. We appreciate this feedback and are pleased that the object-based method and the use of high-resolution simulations were considered valuable contributions.

Specific comments:

I suggest that the authors revise the colormap used in Figure 1a to display better the PV330K field and its characteristics, as the current representation makes the figure difficult to analyze.

Thank you for this suggestion. The color map is designed to highlight the tropopause and allows for easy distinction between stratospheric and tropospheric air. We suggest keeping this as it is established in existing literature (e.g. Oertel et al., 2020).

It could be interesting to add radar-observed images of the event in section 3.

To this end, we will specifically refer to the radar imagery of the event presented in Kopp et al. (2022).

Line 331: I think the authors are referring to the vertical velocity proxy from CAPE ($\sqrt{2CAPE}$).

Well spotted, this will be corrected in the revised manuscript.

**Editor**

Dear Authors,

As you can see the reviewers offer rather different recommendations—Reviewer 1 recommends reject, while Reviewer 2 recommends acceptance pending minor

technical corrections.  In my own reading of the paper, my impression is that the tools and techniques are state-of-the-art and novel (I especially liked the storm-following analyses), and that the paper is well written.

However, I also noted that the paper seems to state a lot of facts about the simulation without a clear story or any in-depth analysis.  This is also reflected, e.g., in the goals of the paper, which promise among other aspects, an analysis of the initiation of convection and of the role of topography, neither of which is meaningfully addressed.  Further, some of the main conclusions are that CAPE decreases along the inflow parcel trajectories, and that inflow air is drawn from south of the Alps.  What is lacking is the an in-depth analysis and the relevance of these findings: Why does CAPE decrease (perhaps you merely sample the updraft as the parcels approach the storm?), and why is it novel/relevant?  Why is the origin of the air south of the Alps relevant?  I also wondered how sensitive the trajectories are e.g., to the output intervals.  Maybe the rogue backward trajectories that terminate south of the Alps result from trajectory errors—backward trajectories in a convergent flow field are prone to interpolation errors.

I think you have all the tools to ask meaningful questions and find some insightful answers.  Perhaps focus on one o two main questions regarding e.g., the environment (storm-centered analyses) or microphysical processes and perform a more in-depth analysis?  See especially the comments by Reviewer 1 who offers a number of suggestions on how to refocus the paper.

I do invite you to revise this paper, but as an option you might also consider withdrawing the current version of the manuscript and to resubmit it once the required, somewhat extensive, revisions are complete.

Thank you for your constructive and detailed feedback on our manuscript, which adds value on top of the reviewers' in-depth comments. We appreciate the opportunity to address the points raised and agree that revisions are necessary to strengthen the scientific contribution of our paper.

You express concerns about the sensitivity of the trajectories and the representativeness of the trajectories with origins south of the Alps. This is a valid point, and accommodations were made during the analysis to ensure the robustness of the results. By starting the trajectories on a fine grid (dx = 250 m) and in high-frequency timesteps (5 min) we essentially produce an ensemble of trajectories, which reflects the inherent uncertainties in the Lagrangian methodology. We will include a discussion of the sensitivity of the trajectories to the output intervals and the potential for interpolation errors in the backward trajectories that terminate south of the Alps, as you suggest.

We appreciate your suggestion to leverage storm-following (object-based) techniques more strongly. Concerning the novelty of the Lagrangian approach, please refer to the general statements made at the beginning of this document.

We acknowledge your observation that the paper currently lacks a cohesive story and in-depth analysis. To address this, we will refocus the manuscript by streamlining the objectives and providing clearer scientific questions that guide the analysis. Specifically, we will rework the introduction and objectives to focus on the environmental conditions (storm-centered analyses) and the physical processes from a Lagrangian perspective, as you suggest. The inclusion of the analysis addressing convection initiation and the role of topography in the introduction was ill-formed and will be revised.

We recognize that several of our conclusions, such as the decrease of CAPE along inflow parcel trajectories and the origin of inflow air south of the Alps, require more discussion and better context to establish their relevance. To address these shortcomings we will further discuss the processes at play (for the decreasing CAPE in the inflowing trajectories you mention, the main effect of the decrease in the final minutes before entering the storm is that they are already vertically displaced, so the portion of CAPE below the parcel is not accessible anymore) and embed our findings into existing severe storm trajectory literature (e.g. Lin and Kumjian, 2022). As mentioned earlier, these changes should also address some of the major comments raised by reviewer 1 about the consideration of existing research into airflow around storms. Further, we will provide a more thorough discussion of the relevance of air originating from south of the Alps and incorporate relevant literature into that argumentation, e.g. Jansing et al. (2024). We will ensure the novelty and implications of these findings are more clearly articulated.

Here are a few specific comments I wrote down while reading the paper:

Line 240: Why not show deep-layer shear (and maybe SRH)? CAPE and shear strongly determine storm structure/intensity, so seeing how they evolve together would be really insightful.

Thank you for this comment. Vertical shear and storm-relative helicity were investigated during the analysis but excluded from the manuscript to focus on other aspects. We will include a discussion of the vertical shear and SRH in the revised manuscript to provide a more comprehensive analysis of the storm environment.

Line 412: Trajectories are indeed very insightful, but in the context of severe storms these have been used for decades, so the novelty is perhaps overstated a bit--or did you mean "storm-following" in this case (object-based)? If so, I agree, and suggest leveraging this method more strongly to present your findings.

Thank you for sharing this concern. It seems that most literature conducting trajectory analysis in convective storms focused on hailstone trajectories rather than air parcel trajectories. As mentioned earlier, to our knowledge, the only paper that focused on air parcel trajectories in severe storms is Lin and Kumjian (2022), who investigated the influences of CAPE on hail production in simulated supercell storms. However, their analysis is performed on an idealized, steady-state storm-centered composite. Thus our work provides insights into the temporal evolution of storms from a — to the best of

our knowledge — novel perspective, and this will be addressed when embedding our methodology into existing literature in the revised manuscript.

Small note, throughout the paper: "Lagrangian trajectories" is redundant: Trajectories always imply that the Lagrangian framework (rather than the Eulerian one) is being used.

Thank you for this comment, it will be addressed in the revised manuscript.

Appendix C has no content.

Thank you for noticing this. Fig. D1 resides in Appendix C; a sentence referencing the figure will be added to Appendix C to avoid the orphan figure / empty section.

**References**

Binder, H., Boettcher, M., Joos, H., and Wernli, H.: The role of warm conveyor belts for the intensification of extratropical cyclones in Northern Hemisphere winter, *J. Atmos. Sci.*, 73, 3997– 4020, https://doi.org/10.1175/JAS-D-15-0302.1, 2016.

Bryan, G. H.; Wyngaard, J. C.; Fritsch, J. M. Resolution Requirements for the Simulation of Deep Moist Convection. *Mon. Weather Rev.*, *131* (10), 2394–2416. https://doi.org/10.1175/1520-0493(2003)131<2394:rrftso>2.0.co;2, 2003.

Catto, J. L., Shaffrey, L. C., and Hodges, K. I.: Can climate models capture the structure of extratropical cyclones*?, J. Climate*, 23,1621–1635, https://doi.org/10.1175/2009JCLI3318.1, 2010.

Cui, R.; Ban, N.; Demory, M.-E.; Aellig, R.; Fuhrer, O.; Jucker, J.; Lapillonne, X.; Schär, C. Exploring Hail and Lightning Diagnostics over the Alpine-Adriatic Region in a Km-Scale Climate Model. *Weather and Climate Dynamics*, *4* (4), 905–926. https://doi.org/10.5194/wcd-4-905-2023, 2023.

Dennis, E. J.; Kumjian, M. R. The Impact of Vertical Wind Shear on Hail Growth in Simulated Supercells. *Journal of the Atmospheric Sciences*, *74* (3), 641–663. https://doi.org/10.1175/JAS-D-16-0066.1, 2017.

Jansing, L.; Papritz, L.; Sprenger, M. A Lagrangian Framework for Detecting and Characterizing the Descent of Foehn from Alpine to Local Scales. *Weather and Climate Dynamics*, *5* (2), 463–489. https://doi.org/10.5194/wcd-5-463-2024, 2024.

Jeevanjee, N. Vertical Velocity in the Gray Zone. *Journal of Advances in Modeling Earth Systems*, *9* (6), 2304–2316. https://doi.org/10.1002/2017MS001059, 2017.

Kopp, J.; Schröer, K.; Schwierz, C.; Hering, A.; Germann, U.; Martius, O. The Summer 2021 Switzerland Hailstorms: Weather Situation, Major Impacts and Unique Observational Data. *Weather*, https://doi.org/10.1002/wea.4306, 2022.

Kumjian, M. R.; Lombardo, K. A Hail Growth Trajectory Model for Exploring the Environmental Controls on Hail Size: Model Physics and Idealized Tests. *J. Atmos. Sci.*, *77* (8), 2765–2791. https://doi.org/10.1175/JAS-D-20-0016.1, 2020.

Lin, Y.; Kumjian, M. R. Influences of CAPE on Hail Production in Simulated Supercell Storms. *J. Atmos. Sci.*, *79* (1), 179–204. https://doi.org/10.1175/JAS-D-21-0054.1, 2022.

Oertel, A.; Boettcher, M.; Joos, H.; Sprenger, M.; Wernli, H. Potential Vorticity Structure of Embedded Convection in a Warm Conveyor Belt and Its Relevance for Large-Scale Dynamics. *Weather Clim. Dynam.*, *1* (1), 127–153. https://doi.org/10.5194/wcd-1-127-2020, 2020.

Prein, A. F.; Liu, C.; Ikeda, K.; Trier, S. B.; Rasmussen, R. M.; Holland, G. J.; Clark, M. P. Increased Rainfall Volume from Future Convective Storms in the US. *Nat. Clim. Change*, *7* (12), 880–884. https://doi.org/10.1038/s41558-017-0007-7, 2017.

Rodwell, M. J.; Wernli, H. Uncertainty Growth and Forecast Reliability during Extratropical Cyclogenesis. *Weather Clim. Dynam., 4* (3), 591–615. https://doi.org/10.5194/wcd-4-591-2023, 2023.

---

## Author Response (AR2)

**Reply document for paper egusphere-2024-2148**

**— round two —**

**An object-based and Lagrangian view on an intense hailstorm day in Switzerland as represented in COSMO-1E ensemble hindcast simulations**

by Killian P. Brennan, Michael Sprenger, André Walser, Marco Arpagaus, and Heini Wernli

March 3, 2025

We thank Reviewer 2 for suggesting to accept the paper as is, and Reviewer 3 for their constructive comments, which helped us to further improve the clarity of our study. In the following, we carefully address all comments of the referees. They are shown in **blue** and our replies in **black**. Mentioned references are listed at the end of the document.
* * *
**Reviewer 2**

**Summary**

I thank the authors for this revision, which has improved it so much. I'm therefore happy to accept the manuscript for publication.

**Reply**: Thank you for your positive evaluation of our manuscript.
* * *
**Reviewer 3**

**Summary**

The authors investigate a severe hailstorm episode over Switzerland, using a high-resolution ensemble. While the analysis is carefully carried out, using suitable tools, I thought the analyses were superficial and did not lead to much new insight. This is reflected by the summary in line 442: As a main result, the authors mention "impressive updraft velocities, overshooting tops, and the intricate cloud structures associated with severe hailstorms (Fig. 5)," while Fig. 5 does not even

show any details (all I see is a coarse outline of a convective cloud). I also have a few comments regarding the interpretation of the results. Overall, my comments probably fall into the major category.

**Reply**: We sincerely appreciate your detailed and insightful review of our manuscript. Your feedback clearly highlighted the importance of a more focused and comprehensive analysis to substantiate our conclusions and to make our conclusions more specific. We genuinely hope that the revised version meets your expectations and that you now regard some of our results as insightful.

Concerning the specific critique about the sentence in the conclusions (L442 in the previous version), we agree that this formulation was not ideal (Fig. 5 does not show "intricate cloud structures") and we improved this formulation as well as a few other formulations in the first part of our conclusions. We however think that, overall, our conclusion provide a balanced view on the novelties of our study (e.g., combined Eulerian and Lagrangian approach, use of ensemble to obtain robust model representations of an observed severe hailstorm) and of its limitations – see in particular the last 3 paragraphs of the paper.

**General comments**

**Reviewer Comment 3.1** — The analysis is based on a convection-allowing simulation, which is understandable because this is what is available operationally. Still, at ∼1 km grid spacing, the convective cells are not well-resolved. I think the authors should acknowledge more clearly that the storm-scale processes are only crudely represented and perhaps resist the temptation to report e.g., updraft speeds and cell sizes (which are almost certainly inaccurately represented).

**Reply 3.1**: Our answer has two parts. We first fully agree with you that 1 km grid spacing is not sufficient to capture all relevant details of deep convective storms and we see a great value in other studies that investigate convective storm dynamics with higher-resolution models. However, to the best of our knowledge, such models are currently not run operationally at numerical weather prediction centers. Our study is part of a project, which is partly driven by stakeholder needs — we would like to use the currently available modeling technology to provide relevant insight into hailstorms in Switzerland. At least in Europe, the model setup we have available (ensemble with 1 km grid spacing and HAILCAST parameterization of hail) is unique and at the forefront of operational NWP. Therefore, in our view, it is still relevant and important to understand how these hailstorms are represented in such a simulation. As this simulation (and models with similar resolutions) are used operationally to forecast hail, it is of value to understand how the model realizes hailstorms, in order to judge whether key processes are adequately accounted for. Additionally, such an analysis can serve as a benchmark for future studies performing hailstorm simulations with improved models (higher resolution and/or improved representation of hail microphysics).

**Reviewer Comment 3.2** — Line 252 ff.: How/where was CAPE and SRH etc. calculated? In the inflow region? How far from the storm? Or is it an average?

**Reply 3.2**: This information is detailed in the caption of Fig. 4. We've improved the description to disambiguate it and it now reads: "The inflow region encompasses a $20 \times 20\,\mathrm{km}^2$ square box, centered 20 km ahead of the storm." To increase readability, we also include this information as a footnote in L252.

**Reviewer Comment 3.3** — When considering the evolution of the CAPE available to a storm centered in the 50x50 grid point subdomain (L345), can you exclude that no other storms are present in the domain? For instance, CAPE might decrease as outflow of neighboring storms enters the domain. So the reported values may not be representative of the CAPE available to the storm in the domain center.

**Reply 3.3**: We appreciate your concern regarding the potential influence of neighboring storms on the CAPE evolution within the $50\times50$ grid point subdomain. As stated above, the inflow region in our analysis encompasses a $20\,\text{km}$ wide box located $10\,\text{km}$ ahead of the storm. Given the average storm motion of approximately $12\,\text{m}\,\text{s}^{-1}$ and the inflow moving towards the storm at $\sim 5\,\text{m}\,\text{s}^{-1}$, the air within this inflow box would reach the storm in roughly 10 minutes.

If a secondary storm were to enter this inflow region and reduce CAPE, it would indeed influence the available energy for the storm under consideration. However, we would argue that this is not a limitation of our approach but rather an inherent feature of real storm interactions, which our analysis aims to capture. Storms in convective environments do not evolve in isolation, and the impact of neighboring convection on CAPE is a relevant dynamical process. Our methodology, therefore, reflects the natural complexity of storm environments rather than imposing an artificial constraint that isolates a single storm from its surroundings.

Moreover, our results remain consistent with expectations based on storm dynamics, suggesting that while external influences such as neighboring convection can locally modulate CAPE, the overall evolution of the storm remains physically interpretable within the framework of our analysis.

We've added the following sentence to L347: "It should be noted here, that neighbouring storms might influence/reduce CAPE in the inflow box."

**Specific comments**

**Reviewer Comment 3.4** — L22: The main characteristic of supercells is their rotating updraft, not necessarily their size (there exist quite small realizations of supercells, e.g., in rainbands of tropical cyclones).

**Reply 3.4**: We've altered the sentence accordingly. The text now reads: "Distinguished by their towering vertical reach spanning the troposphere and rotating updraft, supercells surpass the typical scale of single-cell storms."

**Reviewer Comment 3.5** — L36: Isn't there a distinction between short-term forecasting and nowcasting?

**Reply 3.5**: We agree and did not intend to suggest otherwise. We slightly changed our formulation, which now reads "At short lead times, radar-based nowcasting serves to inform about hail occurrence, where the future state of hailstorms is extrapolated from current radar observations and movement vectors of convective cells (e.g., Hering et al., 2004; Trefalt et al., 2023). Predictions with lead times beyond 1–3 h must rely on numerical weather prediction (NWP) models (e.g., Sun et al., 2014; Nerini et al., 2019)."

**Reviewer Comment 3.6** — L83: The seminal paper on supercell propagation is by Rotunno and Klemp (1982, MWR)

**Reply 3.6**: Thank you, we've included a reference to this seminal paper in our manuscript. The sentence now reads: "Browning (1977) and Rotunno and Klemp (1982) explored storm propagation mechanisms, while Rotunno (1993) analyzed the three-dimensional airflow structure of supercell thunderstorms."

**Reviewer Comment 3.7** — L84: Davies-Jones (1984, JAS) is perhaps more relevant.

**Reply 3.7**: Thank you for mentioning this study, we've included it appropriately. The text now reads: "Despite foundational studies like Browning and Ludlam (1962), Browning (1964), Davies-Jones (1984) that investigated airflow patterns within severe local storms, recent literature predominantly emphasizes hailstone trajectories and over air parcel trajectories in convective systems."

**Reviewer Comment 3.8** — L117: Was ERA5 used for the boundary conditions?

**Reply 3.8**: Information was added accordingly: "The ECMWF global ERA5 reanalysis (Hersbach et al., 2020) was used in this study to characterize the large-scale atmospheric conditions and provided boundary conditions for the regional weather simulations with COSMO."

**Reviewer Comment 3.9** — L123: Is the microphysics scheme single- or double moment? Also, which subgrid-scale turbulence scheme was used?

**Reply 3.9**: Thank you for pointing out this missing information. We've added the corresponding details to the manuscript. The text now reads: "Parameterizations represent unresolved subgrid-scale physical processes, including a single-moment bulk microphysics scheme (Lin et al., 1983) with five species (cloud water, cloud ice, rain, snow, and graupel) and schemes for shallow convection, boundary layer turbulence, radiation, and land-surface processes. The turbulence parameterization is adapted from Mellor and Yamada (1982) with a prognostic equation for the turbulence kinetic energy, including effects from subgrid-scale condensation and evaporation. It is applied to the bottom boundary of the atmospheric model to calculate surface-layer fluxes (Baldauf et al., 2011). The parameterization of radiation follows the scheme of Ritter and Geleyn (1992)."

**Reviewer Comment 3.10** — L145: Why were only the 10+ mm embryos considered?

**Reply 3.10**: In initial tests, we saw that the size distribution of the largest hail embryos was more representative of the radar-observed hail size distribution from the mean diameter over all 5 HAILCAST embryos. Further, the correlation between the largest diameter and the mean across all 5 HAILCAST embryos is high, so most analysis would be interchangeably valid. — No changes were made to the manuscript regarding this comment.

**Reviewer Comment 3.11** — L149: Isn't the main issue that the output time step is rather coarse, rather than the storms being small and moving fast (this is just a matter of domain size and temporal resolution).

**Reply 3.11**: You are right; however, there are technical limits such as storage to take into consideration, which govern the temporal resolution of NWP output in practice. To this end, the

tracking algorithm must account for the limited temporal resolution. And, as far as we know, it is unusual to have 5-min output from an operational NWP ensemble — this was only possible thanks to our colleagues at MeteoSwiss re-running the simulation with "high-frequency" output, which amounted to almost 20 TB.

**Reviewer Comment 3.12**  —  L154: Were these adjustments done manually?

**Reply 3.12**:  The methodology around the adaptive threshold is automated and further described in the appendix (L529 ff.). — No changes were made to the manuscript regarding this comment.

**Reviewer Comment 3.13**  —  L218: Perhaps state the number of storms considered as well?

**Reply 3.13**:  The number of storms is already given at the beginning of Sect. 3.2. We additionally added the number of storms to L217.

**Reviewer Comment 3.14**  —  L238: It is not clear what "the selected storm" refers to, and no specific storm was described earlier (unless I missed it).

**Reply 3.14**:  In L238 we refer to Sect. 3.2 (L222) where we write: *"[...] one exemplary storm was selected from ensemble member 5, which shows a similar realization to the actual storm [...]"*. This storm is highlighted in red in Fig. 3: we've reiterated this in L222 and L238: "(red track in Fig. 3)"

**Reviewer Comment 3.15**  —  L257: Not sure I understand why in your simulation SRH doesn't indicate the potential for storms to rotate. The background value certainly is enhanced within the storm's inflow, but one can still identify a representative "environmental" value that does indicate the supercell potential. Indeed, in Fig. 4, SHR does seem to correlate with the max updraft speed (e.g., increasing at 14:00 UTC).

**Reply 3.15**:  Thank you for pointing out this inaccuracy, we've removed this statement.

**Reviewer Comment 3.16**  —  L273: I suggest removing subjective qualifiers such as "impressive."

**Reply 3.16**:  Done.

**Reviewer Comment 3.17**  —  L287: I'm not sure what you mean by "graupel not activating freezing of cloud water" because cloud ice is present mostly above the homogeneous freezing level; suggest rewording.

**Reply 3.17**:  We've reworded it to make clearer what we mean here. The sentence now reads "However, since the cloud ice is present mostly above the level of homogenous freezing, this suggests that ice introduced by graupel below this level is not activating the freezing of cloud water droplets."

**Reviewer Comment 3.18**  —  L298 ff.: Suggest referring to individual panels in the figure (I wasn't sure if you were talking about panel a or b)

**Reply 3.18**:  Thank you, this is helpful, we added panel specifiers.

**Reviewer Comment 3.19** — L311: I think the size of the storm cores need to be taken with a grain of salt as the storms themselves are very poorly resolved. Yes, you state that at the very end, but why even report numbers if at the end you need to acknowledge that they are likely not accurate.

**Reply 3.19**: In our view, the limitations mentioned in the final section warrant reporting these numbers here. See also our reply to the general comment about model resolution. — No changes were made to the manuscript regarding this comment.

**Reviewer Comment 3.20** — L324: Why does CAPE decrease as the storm passes? (I suspect that increasingly, anvil material is sampled, and at sufficient proximity to the storm, the outflow.) Also, why does CAPE not go to zero? Within the storm, the thermodynamic profile ought to be approximately pseudo-adiabatic.

**Reply 3.20**: The storm consumes some of the CAPE in its environment as it moves air vertically. However, the storm is not a perfect, idealized system — rather, it is a system with limited extent, mixing, etc. — thus, some CAPE remains. If the reviewer knows about an observational study that would reveal that the CAPE decrease should be substantially larger after the passage of the storm, then we would be happy to include it. — No changes were made to the manuscript regarding this comment.

**Reviewer Comment 3.21** — L327: Earlier you stated that the horizontal placement of the hail is not well-represented by the HAILCAST model except for a fudge factor that describes horizontal advection. How confident are you that these results are realistic? If not, why even report them?

**Reply 3.21**: The argumentation provided on L317 ff. corroborate the results on L327: *"Co-location of the hail and updraft maxima is to be expected, as HAILCAST does not account for horizontal advection of hailstones to other grid columns. In contrast, since graupel is explicitly included in the COSMO microphysics, it is subject to horizontal advection in the simulations. Only a small offset of the graupel maximum from the storm center exists (Fig. 6e). The location of graupel gives an upper bound on the potential advection of hail, as graupel has a smaller terminal velocity than even the smallest hailstones, giving more time for horizontal advection to take effect."* We think that this properly describes the limitations when using HAILCAST and why, in this case, we regard the results as still meaningful. — No changes were made to the manuscript regarding this comment.

**Reviewer Comment 3.22** — L338: Suggest flipping Figs. 7a, 7b, and 7c as you first talk about panels b and c, and then panel a.

**Reply 3.22**: Thank you for your suggestion, we've arranged the panels in the order of their references in the text.

**Reviewer Comment 3.23** — L353: You haven't mentioned yet how you calculated the inflow depth. Suggest at least pointing the reader to the trajectories in section 5.

**Reply 3.23**: Thank you for this comment, we've added a reference to Sect. 5 and a footnote reading: "Mean height above ground level of the inflow trajectories in the period $-60$ to $-30$ min before reaching the storm."

**Reviewer Comment 3.24** — L354: Coffer et al. considered the low-level mesocyclone, not the total inflow into the updraft. Suggest e.g., Nowotarski et al. (2020, MWR, "Evaluating the Effective Inflow Layer of Simulated Supercell Updrafts").

**Reply 3.24**: Thank you for pointing this out. We've adjusted the text to: "Throughout the storm's lifetime, the bulk of the inflow originates from 330 – 900 m AGL. Previous studies found the inflow level height of simulated supercells to be between 1400 and 1800 m AGL (Thompson et al., 2007; Nowotarski et al., 2020), while in idealized simulations of supercell low-level mesocyclones, Coffer et al. (2023) found the inflow to be around 200 – 400 m AGL."

**Reviewer Comment 3.25** — L362: You could offer a potential explanation for why $w$ does not seem to be related with CAPE: Vertical pressure-gradient accelerations (e.g., textbook by Markowski and Richardson, 2010, p. 233) and precipitation loading.

**Reply 3.25**: Thank you for this suggestion, we've added a possible explanation to the manuscript.

**Reviewer Comment 3.26** — L376: Were the parcels initialized in a box that had a ceiling of 5000 m AGL (or MSL?)? Or in a layer at 5000 m?

**Reply 3.26**: Thank you for pointing out this ambiguity. The height refers to altitude above mean sea level, we've specified this in the revised manuscript.

**Reviewer Comment 3.27** — L396: In Fig. 10a, it seems that CAPE is pretty steady around 1100 J/kg, so I was unable to see the steady increase mentioned in the text.

**Reply 3.27**: Thank you for pointing this out, we've added the rate of change in CAPE and CIN during this period. Please refer to the track-changes file for the changes made regarding this comment.

**Reviewer Comment 3.28** — L398: Suggest replacing "they" with "the parcels."

**Reply 3.28**: Thank you for pointing out this detail. We've changed this according to your suggestion.

**Reviewer Comment 3.29** — L410: minimum

**Reply 3.29**: Changed as suggested.

**Reviewer Comment 3.30** — L415: I think this is only approximately true (rain falling through an unsaturated parcel breaks the assumption of a closed system, so theta-e would not be perfectly conserved).

**Reply 3.30**: Thank you, we've added an appropriate footnote: "As the trajectories are not a closed system, there might be influence from, e.g., sensible heat transfer from colder rain falling into the warmer air beneath and removing heat energy, effectively reducing $\Theta_e$ in the parcel. However, this effect is expected to be minor, as the temperature difference between the parcel and the infalling rain is limited."

**Reviewer Comment 3.31** — L424: I felt you were overselling the results a bit by claiming that you offered "detailed insights into the dynamic and thermodynamic processes." (You only spent one paragraph on this!)

**Reply 3.31**: **We agree, and we shortened the formulation as follows: "These findings form the basis for the summary and conclusions in the next section."**

**Reviewer Comment 3.32** — Fig. 10: By "outflow" do you mean "anvil outflow"? If so, mention that somewhere, as "outflow" is commonly taken to be the "cool-air outflow" associated with the storm's cold pool.

**Reply 3.32**: Changed as suggested (L412).